# A Fast and Provable Algorithm for Sparse Phase Retrieval

**Jian-Feng Cai**[1,2], **Yu Long**[3*], **Ruixue Wen**[1], **Jiaxi Ying**[1,2*]

[1] Hong Kong University of Science and Technology

[2] HKUST Shenzhen-Hong Kong Collaborative Innovation Research Institute

[3] Guangxi University

## Abstract

We study the sparse phase retrieval problem, which seeks to recover a sparse signal from a limited set of magnitude-only measurements. In contrast to prevalent sparse phase retrieval algorithms that primarily use first-order methods, we propose an innovative second-order algorithm that employs a Newton-type method with hard thresholding. This algorithm overcomes the linear convergence limitations of first-order methods while preserving their hallmark per-iteration computational efficiency. We provide theoretical guarantees that our algorithm converges to the $s$-sparse ground truth signal $\boldsymbol{x}^\natural \in \mathbb{R}^n$ (up to a global sign) at a quadratic convergence rate after at most $O(\log(\|\boldsymbol{x}^\natural\|/x_{\min}^\natural))$ iterations, using $\Omega(s^2 \log n)$ Gaussian random samples. Numerical experiments show that our algorithm achieves a significantly faster convergence rate than state-of-the-art methods.

## 1 Introduction

We study the phase retrieval problem, which involves reconstructing an $n$-dimensional signal $\boldsymbol{x}^\natural$ using its magnitude-only measurements:

$$y_i = |\langle \boldsymbol{a}_i, \boldsymbol{x}^\natural \rangle|^2, \quad i = 1, 2, \cdots, m, \tag{1}$$

where each $y_i$ represents a measurement, $\boldsymbol{a}_i$ denotes a sensing vector, $\boldsymbol{x}^\natural$ is the unknown signal to be recovered, and $m$ is the total number of measurements. The phase retrieval problem arises in various applications, including diffraction imaging (Maiden & Rodenburg, 2009), X-ray crystallography (Miao et al., 2008), and optics (Shechtman et al., 2015).

Although the phase retrieval problem is ill-posed and even NP-hard (Fickus et al., 2014), various algorithms can recover target signals. These are broadly categorized into convex and nonconvex approaches. Convex methods, such as PhaseLift (Candes et al., 2015; 2013), PhaseCut (Waldspurger et al., 2015), and PhaseMax (Goldstein & Studer, 2018; Hand & Voroninski, 2018), offer optimal sample complexity but are computationally challenging in high-dimensional cases. To improve computational efficiency, nonconvex approaches are explored such as alternating minimization (Netrapalli et al., 2013; Cai et al., 2022b), Wirtinger flow (Candes et al., 2015), truncated amplitude flow (Wang et al., 2017a), Riemannian optimization (Cai & Wei, 2023), Gauss-Newton (Gao & Xu, 2017; Ma et al., 2018), Kaczmarz (Wei, 2015; Chi & Lu, 2016), and unregularized gradient descent (Ma et al., 2020). Despite the nonconvex nature of its objective function, the global geometric landscape lacks spurious local minima (Sun et al., 2018; Li et al., 2019; Cai et al., 2023b), allowing algorithms with random initialization to work effectively (Chen et al., 2019; Waldspurger, 2018).

The nonconvex approaches previously mentioned can guarantee successful recovery of the ground truth (up to a global phase) using $m = \Omega(n \log^a n)$ measurements, where $a \geq 0$. This complexity is nearly optimal, as the phase retrieval problem requires $m \geq 2n-1$ for real signals and $m \geq 4n-4$ for complex signals (Conca et al., 2015). However, in practical situations, especially in high-dimensional cases, the number of available measurements is often less than the signal dimension (i.e., $m < n$), leading to a need for further reduction in sample complexity.

---

*Corresponding authors.

In this paper, we focus on the sparse phase retrieval problem, which aims to recover a sparse signal from a limited number of phaseless measurements. It has been established that the minimal sample complexity required to ensure $s$-sparse phase retrievability in the real case is only $2s$ for generic sensing vectors (Wang & Xu, 2014). Several algorithms have been proposed to address the sparse phase retrieval problem (Ohlsson et al., 2012; Cai et al., 2016; Wang et al., 2017b; Jagatap & Hegde, 2019; Cai et al., 2022c). These approaches have been demonstrated to effectively reconstruct the ground truth using $\Omega(s^2 \log n)$ Gaussian measurements. While this complexity is not optimal, it is significantly smaller than that in general phase retrieval.

## 1.1 CONTRIBUTIONS

Existing algorithms for sparse phase retrieval primarily employ first-order methods with linear convergence. Recent work (Cai et al., 2022c) presented a second-order method, while it fails to obtain quadratic convergence. The main contributions of this paper can be summarized in three points:

1. We propose a second-order algorithm for sparse phase retrieval that maintains the same per-iteration computational complexity as popular first-order methods. Our algorithm enhances convergence by integrating second-order derivative information from intensity-based empirical loss into the search direction. To ensure computational efficiency, the Newton step is applied only to a subset of variables identified through the amplitude-based empirical loss.

2. We establish a non-asymptotic quadratic convergence rate for our proposed algorithm and provide the iteration complexity. Specifically, we prove that the algorithm converges to the ground truth (up to a global sign) at a quadratic rate after at most $O(\log(\|\boldsymbol{x}^\natural\|/x_{\min}^\natural))$ iterations, with $m = \Omega(s^2 \log n)$ measurements. To the best of our knowledge, this is the first algorithm to establish a quadratic convergence rate for sparse phase retrieval.

3. Numerical experiments demonstrate that the proposed algorithm achieves a significantly faster convergence rate in comparison to state-of-the-art methods. The experiments also reveal that our algorithm attains higher success rates in the exact recovery of signals from noise-free measurements and offers improved signal reconstruction in the presence of noise.[1]

**Notation:** $\|\boldsymbol{x}\|_0$ denotes the number of nonzero entries of $\boldsymbol{x}$, and $\|\boldsymbol{x}\|$ denotes the $\ell_2$-norm. For a matrix $\boldsymbol{A} \in \mathbb{R}^{m \times n}$, $\|\boldsymbol{A}\|$ is the spectral norm of $\boldsymbol{A}$. For any $q_1 \geq 1$ and $q_2 \geq 1$, $\|\boldsymbol{A}\|_{q_2 \to q_1}$ denotes the induced operator norm from the Banach space $(\mathbb{R}^n, \|\cdot\|_{q_2})$ to $(\mathbb{R}^m, \|\cdot\|_{q_1})$. $\lambda_{\min}(\boldsymbol{A})$ and $\lambda_{\max}(\boldsymbol{A})$ denote the smallest and largest eigenvalues of $\boldsymbol{A}$. $|\mathcal{S}|$ denotes the number of elements in $\mathcal{S}$. $\boldsymbol{a} \odot \boldsymbol{b}$ denotes the entrywise product of $\boldsymbol{a}$ and $\boldsymbol{b}$. We write $f(n) = O(g(n))$ to indicate $f(n) \leq c_1 g(n)$ for some constant $c_1 > 0$, and $f(n) = \Omega(g(n))$ to denote $f(n) \geq c_2 g(n)$ for some constant $c_2 > 0$. For $\boldsymbol{x}, \boldsymbol{x}^\natural \in \mathbb{R}^n$, the distance between $\boldsymbol{x}$ and $\boldsymbol{x}^\natural$ is defined as $\operatorname{dist}(\boldsymbol{x}, \boldsymbol{x}^\natural) := \min\left\{\|\boldsymbol{x} - \boldsymbol{x}^\natural\|, \|\boldsymbol{x} + \boldsymbol{x}^\natural\|\right\}$. $x_{\min}^\natural$ denotes the smallest nonzero entry in magnitude of $\boldsymbol{x}^\natural$.

## 2 PROBLEM FORMULATION AND RELATED WORK

We first present the problem formulation for sparse phase retrieval, and then review related work and provide a comparative overview of state-of-the-art algorithms and our proposed algorithm.

### 2.1 PROBLEM FORMULATION

The standard sparse phase retrieval problem can be concisely expressed as finding $\boldsymbol{x}$ that satisfies

$$|\langle \boldsymbol{a}_i, \boldsymbol{x} \rangle|^2 = y_i \quad \forall i = 1, \ldots, m, \quad \text{and} \quad \|\boldsymbol{x}\|_0 \leq s, \tag{2}$$

where $\{\boldsymbol{a}_i\}_{i=1}^m$ and $\{y_i\}_{i=1}^m$ represent known and fixed sensing vectors and phaseless measurements, respectively. Each $y_i = |\langle \boldsymbol{a}_i, \boldsymbol{x}^\natural \rangle|^2$ with $\boldsymbol{x}^\natural$ the ground truth ($\|\boldsymbol{x}^\natural\|_0 \leq s$). While sparsity level is assumed known a priori for theoretical analysis, our experiments will explore cases with unknown $s$. To address Problem (2), various problem reformulations have been explored. Convex formulations, such as the $\ell_1$-regularized PhaseLift method (Ohlsson et al., 2012), often use the lifting technique and solve the problem in the $n \times n$ matrix space, resulting in high computational costs.

---

[1]Our codes are available at https://github.com/jxying/SparsePR.

Table 1: Overview of per-iteration computational cost, iteration complexity, and loss function types for ThWF (Cai et al., 2016), SPARTA (Wang et al., 2017b), CoPRAM (Jagatap & Hegde, 2019), HTP (Cai et al., 2022c), and our proposed algorithm. Here, $x^\natural$ represents the ground truth with dimension $n$ and sparsity $s$, and $x^\natural_{\min}$ denotes the smallest nonzero entry in magnitude of $x^\natural$.

| Methods | Per-iteration computation | Iteration complexity | Loss function |
|---------|---------------------------|----------------------|---------------|
| ThWF | $O(n^2 \log n)$ | $O(\log(1/\epsilon))$ | $f_I(\boldsymbol{x})$ |
| SPARTA | $O(ns^2 \log n)$ | $O(\log(1/\epsilon))$ | $f_A(\boldsymbol{x})$ |
| CoPRAM | $O(ns^2 \log n)$ | $O(\log(1/\epsilon))$ | $f_A(\boldsymbol{x})$ |
| HTP | $O((n+s^2)s^2 \log n)$ | $O(\log(\log(n^{s^2})) + \log(\|\boldsymbol{x}^\natural\|/x^\natural_{\min}))$ | $f_A(\boldsymbol{x})$ |
| Proposed | $O((n+s^2)s^2 \log n)$ | $O(\log(\log(1/\epsilon)) + \log(\|\boldsymbol{x}^\natural\|/x^\natural_{\min}))$ | $f_I(\boldsymbol{x}), f_A(\boldsymbol{x})$ |

To enhance computational efficiency, nonconvex approaches (Cai et al., 2016; Wang et al., 2017b; Cai et al., 2022c; Soltanolkotabi, 2019) are explored, which can be formulated as:

$$\underset{\boldsymbol{x}}{\text{minimize}} \ f(\boldsymbol{x}), \qquad \text{subject to} \quad \|\boldsymbol{x}\|_0 \leq s. \tag{3}$$

Both the loss function $f(\boldsymbol{x})$ and the $\ell_0$-norm constraint in Problem (3) are nonconvex, making it challenging to solve. Two prevalent loss functions are investigated: intensity-based empirical loss

$$f_I(\boldsymbol{x}) := \frac{1}{4m} \sum_{i=1}^{m} \left( |\langle \boldsymbol{a}_i, \boldsymbol{x} \rangle|^2 - y_i \right)^2, \tag{4}$$

and amplitude-based empirical loss

$$f_A(\boldsymbol{x}) := \frac{1}{2m} \sum_{i=1}^{m} \left( |\langle \boldsymbol{a}_i, \boldsymbol{x} \rangle| - z_i \right)^2, \tag{5}$$

where $z_i = \sqrt{y_i}$, $i = 1, \ldots, m$. The intensity-based loss $f_I(\boldsymbol{x})$ is smooth, while the amplitude-based loss $f_A(\boldsymbol{x})$ is non-smooth because of the modulus.

## 2.2 RELATED WORK

Existing nonconvex sparse phase retrieval algorithms can be broadly classified into two categories: gradient projection methods and alternating minimization methods. Gradient projection methods, such as ThWF (Cai et al., 2016) and SPARTA (Wang et al., 2017b), employ thresholded gradient descent and iterative hard thresholding, respectively. On the other hand, alternating minimization methods, including CoPRAM (Jagatap & Hegde, 2019) and HTP (Cai et al., 2022c), alternate between updating the signal and phase. When updating the signal, formulated as a sparsity-constrained least squares problem, CoPRAM leverages the cosamp method (Needell & Tropp, 2009), while HTP applies the hard thresholding pursuit algorithm (Foucart, 2011).

Contrary to previously discussed algorithms, our algorithm is rooted in a Newton-type method with hard thresholding and incorporates second-order information to accelerate convergence. Unlike alternating minimization methods, our algorithm eliminates the need to update the signal and phase separately. A sample complexity of $\Omega(s^2 \log n)$ under Gaussian measurements is required for successful recovery across all discussed algorithms. ThWF employs an intensity-based loss as the objective function, while SPARTA, CoPRAM, and HTP utilize an amplitude-based loss. Our algorithm, distinctively, adopts both loss types: It uses intensity-based loss as the objective function and amplitude-based loss to determine the support for the Newton update.

**Discussions:** Table 1 provides a comparative overview of various algorithms. ThWF, SPARTA, and CoPRAM, as first-order methods, exhibit linear convergence. In contrast, our algorithm achieves a quadratic convergence rate. Although HTP is a second-order method that converges in a finite number of iterations, it fails to establish quadratic convergence and has a higher iteration complexity than our algorithm. Empirical evidence further shows that our algorithm converges faster than HTP. This can be attributed to our algorithm's more effective exploitation of the second-order information of the objective function when constructing the search direction.

## 3    MAIN RESULTS

In this section, we present our proposed algorithm for sparse phase retrieval. Generally, nonconvex methods comprise two stages: initialization and refinement. The first stage generates an initial guess close to the target signal, while the second stage refines the initial guess. Our proposed algorithm adheres to this two-stage strategy. In the first stage, we employ an existing effective method to generate an initial point. Our primary focus is on the second stage, wherein we propose an efficient second-order algorithm to refine the initial guess.

### 3.1    PROPOSED ALGORITHM

We introduce our proposed second-order algorithm for sparse phase retrieval, which adopts a dual loss strategy: It employs the intensity-based loss as defined in (4) for the objective function and uses the amplitude-based loss from (5) to identify the support for the Newton update. In alignment with established Newton-type methods for sparse optimization (Cai et al., 2023a; Zhou et al., 2021; Meng & Zhao, 2020), our algorithm involves two primary steps: (1) identifying the sets of *free* and *fixed* variables, and (2) computing the search direction.

#### 3.1.1    IDENTIFYING FREE AND FIXED VARIABLES

In Newton-type methods, computing the Newton direction at each iteration typically requires solving a linear system, a process that often incurs a significant computational cost of $O(n^\omega)$. Here, $n$ denotes the dimension of the problem space, and $\omega$ represents the matrix multiplication constant, which could be less than 3 if fast matrix multiplication is employed. Unfortunately, this complexity can still render the algorithm impractical for high-dimensional scenarios where $n$ is large.

To address this challenge, we categorize variables into two groups at each iteration: *free* and *fixed*, and update them separately. The *free* variables, which consist of at most $s$ variables, are updated according to the (approximate) Newton direction, while the *fixed* variables are set to zero. This strategy substantially cuts the computational cost from $O(n^\omega)$ to $O(s^\omega)$ by only requiring the solution of a linear system of size $s \times s$, and ensures $s$-sparsity at each iteration.

We identify *free* variables using one-step iterative hard thresholding (IHT) of the loss $f_A(\boldsymbol{x})$ in (5):

$$\mathcal{S}_{k+1} = \text{supp}\left(\mathcal{H}_s(\boldsymbol{x}^k - \eta \nabla f_A(\boldsymbol{x}^k))\right),$$

where $\eta$ is the stepsize, and $\mathcal{H}_s$ denotes the $s$-sparse hard-thresholding operator, defined as follows:

$$\mathcal{H}_s(\boldsymbol{w}) := \arg\min_{\boldsymbol{x}} \|\boldsymbol{x} - \boldsymbol{w}\|^2, \quad \text{subject to} \quad \|\boldsymbol{x}\|_0 \leq s, \tag{6}$$

This operator picks the $s$ largest magnitude entries of the input vector and sets the rest to zero. Therefore, there are at most $s$ *free* variables. Since $f_A$ is non-smooth, we adopt the generalized gradient (Zhang et al., 2017) as $\nabla f_A$. The set of *fixed* variables is the complement of $\mathcal{S}_{k+1}$. We only update *free* variables along the approximate Newton direction and set others to zero.

While the objective function of our algorithm is the intensity-based loss $f_I(\boldsymbol{x})$, it's noteworthy that we use the amplitude-based loss $f_A(\boldsymbol{x})$ in place of $f_I(\boldsymbol{x})$ when identifying *free* variables. Empirical evidence shows that this approach yields faster convergence compared to using $f_I(\boldsymbol{x})$.

#### 3.1.2    COMPUTING SEARCH DIRECTION

We update the *free* variables in $\mathcal{S}_{k+1}$ by solving the following support-constrained problem:

$$\underset{\boldsymbol{x}}{\text{minimize}}\ \psi_k(\boldsymbol{x}), \quad \text{subject to} \quad \text{supp}(\boldsymbol{x}) \subseteq \mathcal{S}_{k+1}, \tag{7}$$

where $\psi_k(\boldsymbol{x})$ is an approximation of the intensity-based loss $f_I$ at $\boldsymbol{x}^k$. To ensure fast convergence and efficient computation, we choose $\psi_k(\boldsymbol{x})$ in (7) as the second-order Taylor expansion of $f_I$ at $\boldsymbol{x}^k$:

$$\psi_k(\boldsymbol{x}) := f_I(\boldsymbol{x}^k) + \langle \nabla f_I(\boldsymbol{x}^k),\ \boldsymbol{x} - \boldsymbol{x}^k \rangle + \frac{1}{2}\langle \boldsymbol{x} - \boldsymbol{x}^k,\ \nabla^2 f_I(\boldsymbol{x}^k)(\boldsymbol{x} - \boldsymbol{x}^k) \rangle.$$

For notational simplicity, define $\boldsymbol{g}_{\mathcal{S}_{k+1}}^k = \left[\nabla f_I(\boldsymbol{x}^k)\right]_{\mathcal{S}_{k+1}}$, which denotes the sub-vector of $\nabla f_I(\boldsymbol{x}^k)$ indexed by $\mathcal{S}_{k+1}$, $\boldsymbol{H}_{\mathcal{S}_{k+1},\mathcal{S}_{k+1}}^k = \left[\nabla^2 f_I(\boldsymbol{x}^k)\right]_{\mathcal{S}_{k+1},\mathcal{S}_{k+1}}$, which represents the principle sub-matrix

of the Hessian indexed by $\mathcal{S}_{k+1}$, and $\boldsymbol{H}^k_{\mathcal{S}_{k+1}, \mathcal{S}^c_{k+1}} = \left[\nabla^2 f_I(\boldsymbol{x}^k)\right]_{\mathcal{S}_{k+1}, \mathcal{S}^c_{k+1}}$, denoting the sub-matrix whose rows and columns are indexed by $\mathcal{S}_{k+1}$ and $\mathcal{S}^c_{k+1}$, respectively. Let $\boldsymbol{x}^\star$ denote the minimizer of Problem (7). Following from the first-order optimality condition of Problem (7), we obtain that $\boldsymbol{x}^\star_{\mathcal{S}^c_{k+1}} = \boldsymbol{0}$ and $\boldsymbol{x}^\star_{\mathcal{S}_{k+1}}$ satisfies

$$\boldsymbol{H}^k_{\mathcal{S}_{k+1}, \mathcal{S}_{k+1}}\left(\boldsymbol{x}^\star_{\mathcal{S}_{k+1}} - \boldsymbol{x}^k_{\mathcal{S}_{k+1}}\right) = \boldsymbol{H}^k_{\mathcal{S}_{k+1}, \mathcal{S}^c_{k+1}}\boldsymbol{x}^k_{\mathcal{S}^c_{k+1}} - \boldsymbol{g}^k_{\mathcal{S}_{k+1}}. \tag{8}$$

As a result, we obtain the next iterate $\boldsymbol{x}^{k+1}$ by

$$\boldsymbol{x}^{k+1}_{\mathcal{S}_{k+1}} = \boldsymbol{x}^k_{\mathcal{S}_{k+1}} - \boldsymbol{p}^k_{\mathcal{S}_{k+1}}, \quad \text{and} \quad \boldsymbol{x}^{k+1}_{\mathcal{S}^c_{k+1}} = \boldsymbol{0}, \tag{9}$$

where $\boldsymbol{p}^k_{\mathcal{S}_{k+1}}$ represents the approximate Newton direction over $\mathcal{S}_{k+1}$, which can be calculated by

$$\boldsymbol{H}^k_{\mathcal{S}_{k+1}, \mathcal{S}_{k+1}}\boldsymbol{p}^k_{\mathcal{S}_{k+1}} = -\boldsymbol{H}^k_{\mathcal{S}_{k+1}, J_{k+1}}\boldsymbol{x}^k_{J_{k+1}} + \boldsymbol{g}^k_{\mathcal{S}_{k+1}}. \tag{10}$$

where $J_{k+1} := \mathcal{S}_k \setminus \mathcal{S}_{k+1}$ with $|J_{k+1}| \leq s$. In contrast to Eq. (8), we replace $\boldsymbol{x}^k_{\mathcal{S}^c_{k+1}}$ with $\boldsymbol{x}^k_{J_{k+1}}$ in (10), as $J_{k+1}$ captures all nonzero elements in $\boldsymbol{x}^k_{\mathcal{S}^c_{k+1}}$ as follows:

$$\mathcal{G}\left(\boldsymbol{x}^k_{\mathcal{S}^c_{k+1}}\right) = \begin{bmatrix} \boldsymbol{x}^k_{\mathcal{S}^c_{k+1} \cap \mathcal{S}_k} \\ \boldsymbol{0} \end{bmatrix} = \begin{bmatrix} \boldsymbol{x}^k_{\mathcal{S}_k \setminus \mathcal{S}_{k+1}} \\ \boldsymbol{0} \end{bmatrix} = \begin{bmatrix} \boldsymbol{x}^k_{J_{k+1}} \\ \boldsymbol{0} \end{bmatrix}, \tag{11}$$

where operator $\mathcal{G}$ arranges all nonzero elements of a vector to appear first, followed by zero elements. The first equality in (11) follows from the fact that $\text{supp}(\boldsymbol{x}^k) \subseteq \mathcal{S}_k$.

By calculating $\boldsymbol{H}^k_{\mathcal{S}_{k+1}, J_{k+1}}$ rather than $\boldsymbol{H}^k_{\mathcal{S}_{k+1}, \mathcal{S}^c_{k+1}}$ as in (10), the computational cost is substantially reduced from $O(smn)$ to $O(s^2m)$. The costs for computing $\boldsymbol{H}^k_{\mathcal{S}_{k+1}, \mathcal{S}_{k+1}}$ and solving the linear system in (10) are $O(s^2m)$ and $O(s^\omega)$, respectively. Thus, the cost for Step 2 is $O(s^2m)$. The cost for Step 1 amounts to $O(mn)$, which involves calculating $\nabla f_A(\boldsymbol{x}^k)$. Therefore, the overall cost per iteration is $O(n + s^2)m$, with $m$ on the order of $s^2 \log n$, which is generally necessary for a theoretical recovery guarantee. Assuming $s = O(\sqrt{n})$ (otherwise, the complexity $\Omega(s^2 \log n)$ for sparse phase retrieval would reduce to that of general methods), our algorithm's per-iteration complexity matches that of popular first-order methods at $O(ns^2 \log n)$.

---

**Algorithm 1** Proposed algorithm

---

**Input:** Data $\{\boldsymbol{a}_i, y_i\}^m_{i=1}$, sparsity $s$, initial estimate $\boldsymbol{x}^0$, and stepsize $\eta$.
1: **for** $k = 0, 1, 2, \ldots$ **do**
2:   Identify the set of *free* variables $\mathcal{S}_{k+1} = \text{supp}(\mathcal{H}_s(\boldsymbol{x}^k - \eta \nabla f_A(\boldsymbol{x}^k)))$;
3:   Compute the search direction $\boldsymbol{p}^k_{\mathcal{S}_{k+1}}$ over $\mathcal{S}_{k+1}$ by solving (10);
4:   Update $\boldsymbol{x}^{k+1}$:
$$\boldsymbol{x}^{k+1}_{\mathcal{S}_{k+1}} = \boldsymbol{x}^k_{\mathcal{S}_{k+1}} - \boldsymbol{p}^k_{\mathcal{S}_{k+1}}, \quad \text{and} \quad \boldsymbol{x}^{k+1}_{\mathcal{S}^c_{k+1}} = \boldsymbol{0}.$$
5: **end for**
**Output:** $\boldsymbol{x}^{k+1}$.

---

### 3.1.3 LEVERAGING TWO TYPES OF LOSSES

We provide an intuitive explanation of our algorithm, with a particular focus on the utilization of two types of loss functions: the intensity-based loss and the amplitude-based loss.

Our algorithm employs the intensity-based loss $f_I$ as the objective function; its smoothness facilitates the computation of the Hessian and the construction of the Newton direction. While one might intuitively use the same loss function to determine variables for the Newton update, our numerical experiments indicate that the amplitude-based loss $f_A$ is more effective for this purpose.

One potential reason is the superior curvature exhibited by $f_A$ around the true solution, which often behaves more similarly to that of a quadratic least-squares loss (Chi et al., 2019; Zhang et al., 2017; Wang et al., 2017a). This indicates that the gradient of $f_A$ could offer a more effective search direction than that of $f_I$. Furthermore, studies (Zhang et al., 2018; Cai et al., 2022a) have demonstrated that algorithms founded on $f_A$ consistently outperform those based on $f_I$ in numerical results.

## 3.2 INITIALIZATION

The nonconvex nature of phase retrieval problems often requires a good initial guess to find the target signal. Spectral initialization (Candes et al., 2015) is a common approach. We adopt a sparse variant of the spectral initialization method to obtain a favorable initial guess. It would be intriguing to consider other well-established initialization methods, such as sparse orthogonality-promoting initialization (Wang et al., 2017b), diagonal thresholding initialization (Cai et al., 2016), and modified spectral initialization method (Cai et al., 2023c).

Assuming $\{\boldsymbol{a}_i\}_{i=1}^m$ are independently drawn from a Gaussian distribution $\mathcal{N}(\boldsymbol{0}, \boldsymbol{I}_n)$, the expectation of the matrix $\frac{1}{m}\sum_{i=1}^m y_i \boldsymbol{a}_i \boldsymbol{a}_i^T$ is $\boldsymbol{M} := \|\boldsymbol{x}^\natural\|^2 \boldsymbol{I}_n + 2\boldsymbol{x}^\natural(\boldsymbol{x}^\natural)^T$. The leading eigenvector of $\boldsymbol{M}$ is precisely $\pm\boldsymbol{x}^\natural$. Hence, the leading eigenvector of $\frac{1}{m}\sum_{i=1}^m y_i \boldsymbol{a}_i \boldsymbol{a}_i^T$ can be close to $\pm\boldsymbol{x}^\natural$ (Candes et al., 2015). However, this method requires the sample complexity of at least $\Omega(n)$, which is excessively high for sparse phase retrieval. Leveraging the sparsity of $\boldsymbol{x}^\natural$ is crucial to lower this complexity.

We adopt the sparse spectral initialization method proposed in (Jagatap & Hegde, 2019). Specifically, we first collect the indices of the largest $s$ values from $\left\{\frac{1}{m}\sum_{i=1}^m y_i[\boldsymbol{a}_i]_j^2\right\}_{j=1}^n$ and obtain the set $\hat{S}$, serving as an estimate of the support of the true signal $\boldsymbol{x}^\natural$. Next, we construct the initial guess $\boldsymbol{x}^0$ as follows: $\boldsymbol{x}_{\hat{S}}^0$ is the leading eigenvector of $\frac{1}{m}\sum_{i=1}^m y_i[\boldsymbol{a}_i]_{\hat{S}}[\boldsymbol{a}_i]_{\hat{S}}^T$, and $\boldsymbol{x}_{\hat{S}^c}^0 = \boldsymbol{0}$. Finally, we scale $\boldsymbol{x}^0$ such that $\|\boldsymbol{x}^0\|^2 = \frac{1}{m}\sum_{i=1}^m y_i$, ensuring the power of $\boldsymbol{x}^0$ closely aligns with the power of $\boldsymbol{x}^\natural$.

The study in (Jagatap & Hegde, 2019) demonstrates that, given a sample complexity $m = \Omega(s^2 \log n)$, the aforementioned sparse spectral initialization method can produce an initial estimate $\boldsymbol{x}^0$ that falls within a small-sized neighborhood around the ground truth. Specifically, it holds $\mathrm{dist}(\boldsymbol{x}^0, \boldsymbol{x}^\natural) \leq \gamma\|\boldsymbol{x}^\natural\|$ for any $\gamma \in (0, 1)$, with a probability of at least $1 - 8m^{-1}$. Our theoretical analysis demonstrates that, once the estimate enters this neighborhood, our iterative updates in the refinement stage will consistently stay within this neighborhood and be attracted towards the ground truth, exhibiting at least linear convergence and achieving quadratic convergence after $K$ iterations.

## 3.3 THEORETICAL RESULTS

Given the nonconvex nature of the objective function and constraint set in the sparse phase retrieval problem, a theoretical analysis is crucial to guarantee the convergence of our algorithm. In this subsection, we provide a comprehensive convergence analysis for both noise-free and noisy cases.

### 3.3.1 NOISE-FREE CASE

We begin by the noise-free case, in which each measurement $y_i = |\langle \boldsymbol{a}_i, \boldsymbol{x}^\natural\rangle|^2$. Starting with an initial guess obtained via the sparse spectral initialization method, the following theorem shows that our algorithm exhibits a quadratic convergence rate after at most $O(\log(\|\boldsymbol{x}^\natural\|/x_{\min}^\natural))$ iterations.

**Theorem 3.1.** *Let $\{\boldsymbol{a}_i\}_{i=1}^m$ be i.i.d. random vectors distributed as $\mathcal{N}(\boldsymbol{0}, \boldsymbol{I}_n)$, and $\boldsymbol{x}^\natural \in \mathbb{R}^n$ be any signal with $\|\boldsymbol{x}^\natural\|_0 \leq s$. Let $\{\boldsymbol{x}^k\}_{k\geq 1}$ be the sequence generated by Algorithm 1 with the input measurements $y_i = |\langle \boldsymbol{a}_i, \boldsymbol{x}^\natural\rangle|^2$, $i = 1, \ldots, m$, and the initial guess $\boldsymbol{x}^0$ generated by the sparse spectral initialization method mentioned earlier. There exist positive constants $\rho, \eta_1, \eta_2, C_1, C_2, C_3, C_4, C_5$ such that if the stepsize $\eta \in [\eta_1, \eta_2]$ and $m \geq C_1 s^2 \log n$, then with probability at least $1 - (C_2 K + C_3)m^{-1}$, the sequence $\{\boldsymbol{x}^k\}_{k\geq 1}$ converges to the ground truth $\boldsymbol{x}^\natural$ at a quadratic rate after at most $O\big(\log(\|\boldsymbol{x}^\natural\|/x_{\min}^\natural)\big)$ iterations, i.e.,*

$$\mathrm{dist}(\boldsymbol{x}^{k+1}, \boldsymbol{x}^\natural) \leq \rho \cdot \mathrm{dist}^2(\boldsymbol{x}^k, \boldsymbol{x}^\natural), \quad \forall\, k \geq K,$$

*where $K \leq C_4 \log\big(\|\boldsymbol{x}^\natural\|/x_{\min}^\natural\big) + C_5$, and $x_{\min}^\natural$ is the smallest nonzero entry in magnitude of $\boldsymbol{x}^\natural$.*

The proof of Theorem 3.1 is available in Appendix B.3.1. Theorem 3.1 establishes the non-asymptotic quadratic convergence rate of our algorithm as it converges to the ground truth, leading to an iteration complexity of $O(\log(\log(1/\epsilon)) + \log(\|\boldsymbol{x}^\natural\|/x_{\min}^\natural))$ for achieving an $\epsilon$-accurate solution. While superlinear convergence for Newton-type methods has been widely recognized in literature, it typically holds only asymptotically. Consequently, the actual iteration complexity remains undefined, highlighting the importance of establishing a non-asymptotic superlinear convergence rate. For the first $K$ iterations, our algorithm exhibits at least linear convergence.

### 3.3.2 Noisy case

We demonstrate the robustness of our algorithm under noisy measurement conditions. Building upon (Cai et al., 2016; Chen & Candès, 2017), we assume that the noisy measurements are given by:

$$y_i = |\langle \boldsymbol{a}_i, \boldsymbol{x}^\natural \rangle|^2 + \epsilon_i, \quad \text{for } i = 1, \ldots, m,$$

where $\boldsymbol{\epsilon}$ represents a vector of stochastic noise that is independent of $\{\boldsymbol{a}_i\}_{i=1}^m$. Throughout this paper, we assume, without loss of generality, that the expected value of $\boldsymbol{\epsilon}$ is $\boldsymbol{0}$.

**Theorem 3.2.** *Let $\{\boldsymbol{a}_i\}_{i=1}^m$ be i.i.d. random vectors distributed as $\mathcal{N}(\boldsymbol{0}, \boldsymbol{I}_n)$, and $\boldsymbol{x}^\natural \in \mathbb{R}^n$ be any signal with $\|\boldsymbol{x}^\natural\|_0 \leq s$. Let $\{\boldsymbol{x}^k\}_{k \geq 1}$ be the sequence generated by Algorithm 1 with noisy input $y_i = |\langle \boldsymbol{a}_i, \boldsymbol{x}^\natural \rangle|^2 + \epsilon_i$, $i = 1, \ldots, m$. There exists positive constants $\eta_1, \eta_2, C_6, C_7, C_8$, and $\gamma \in (0, 1/8]$, such that if the stepsize $\eta \in [\eta_1, \eta_2]$, $m \geq C_6 s^2 \log n$ and the initial guess $\boldsymbol{x}^0$ obeys $\mathrm{dist}(\boldsymbol{x}^0, \boldsymbol{x}^\natural) \leq \gamma \|\boldsymbol{x}^\natural\|$ with $\|\boldsymbol{x}^0\|_0 \leq s$, then with probability at least $1 - (C_7 K' + C_8) m^{-1}$,*

$$\mathrm{dist}(\boldsymbol{x}^{k+1}, \boldsymbol{x}^\natural) \leq \rho' \cdot \mathrm{dist}(\boldsymbol{x}^k, \boldsymbol{x}^\natural) + \upsilon \|\boldsymbol{\epsilon}\|, \quad \forall 0 \leq k \leq K',$$

*where $\rho' \in (0, 1)$, $\upsilon \in (0, 1)$, and $K'$ is a positive integer.*

The proof of Theorem 3.2 is provided in Appendix B.3.2. Theorem 3.2 validates the robustness of our algorithm, demonstrating its ability to effectively recover the signal from noisy measurements. It reveals a linear convergence rate for our algorithm in the presence of noise. However, our algorithm incorporates the second-order information when determining the search direction, which always leads to faster convergence than first-order algorithms. In addition, Theorem 3.2 does not suggest that the established inequality fails to hold when $k \geq K'$.

## 4 Experimental Results

In this section, we present a series of numerical experiments designed to validate the efficiency and accuracy of our proposed algorithm. All experiments were conducted on a 2 GHz Intel Core i5 processor with 16 GB of RAM, and all compared methods were implemented using MATLAB.

Unless explicitly specified, the sensing vectors $\{\boldsymbol{a}_i\}_{i=1}^m$ were generated by the standard Gaussian distribution. The true signal $\boldsymbol{x}^\natural$ has $s$ nonzero entries, where the support is selected uniformly from all subsets of $[n]$ with cardinality $s$, and their values are independently generated from the standard Gaussian distribution $\mathcal{N}(0, 1)$. In the case of noisy measurements, we have:

$$y_i = |\langle \boldsymbol{a}_i, \boldsymbol{x}^\natural \rangle|^2 + \sigma \varepsilon_i, \quad \text{for } i = 1, \ldots, m, \tag{12}$$

where $\{\varepsilon_i\}_{i=1}^m$ follow i.i.d standard Gaussian distribution, and $\sigma > 0$ determines the noise level.

We compare the convergence speed of our algorithm with state-of-the-art methods, including ThWF (Cai et al., 2016), SPARTA (Wang et al., 2017b), CoPRAM (Jagatap & Hegde, 2019), and HTP (Cai et al., 2022c). We fine-tune the parameters and set: $\alpha = 0.7$ for ThWF; $\gamma = 0.5$, $\mu = 1$ and $|\mathcal{I}| = \lceil m/6 \rceil$ for SPARTA; $\eta = 0.95$ for both HTP and our algorithm. The maximum number of iterations for each algorithm is 1000. The Relative Error (RE) between the estimated signal $\hat{\boldsymbol{x}}$ and the ground truth $\boldsymbol{x}^\natural$ is defined as $\mathrm{RE} := \frac{\mathrm{dist}(\hat{\boldsymbol{x}}, \boldsymbol{x}^\natural)}{\|\boldsymbol{x}^\natural\|}$. A recovery is deemed successful if $\mathrm{RE} < 10^{-3}$.

Figure 1 compares the number of iterations required for convergence across various algorithms. We set the number of measurements to $m = 3000$, the dimension of the signal to $n = 5000$, and the sparsity levels to $s = 80$ and $100$. We consider both noise-free and noisy measurements with a noise level of $\sigma = 0.03$. We observe that all algorithms perform well under both noise-free and noisy conditions; however, our algorithm converges with significantly fewer iterations.

Table 2 presents a comparison of the convergence running times for various algorithms, corresponding to the experiments depicted in Figure 1. For noise-free measurements, all algorithms are set to terminate when the iterate satisfies the following condition: $\frac{\mathrm{dist}(\boldsymbol{x}^k, \boldsymbol{x}^\natural)}{\|\boldsymbol{x}^\natural\|} < 10^{-3}$, which indicates a successful recovery. In the case of noisy measurements, the termination criterion is set as $\frac{\mathrm{dist}(\boldsymbol{x}^{k+1}, \boldsymbol{x}^k)}{\|\boldsymbol{x}^k\|} < 10^{-3}$. As evidenced by the results in Table 2, our algorithm consistently outperforms state-of-the-art methods in terms of running time, for both noise-free and noisy cases, highlighting its superior efficiency for sparse phase retrieval applications.

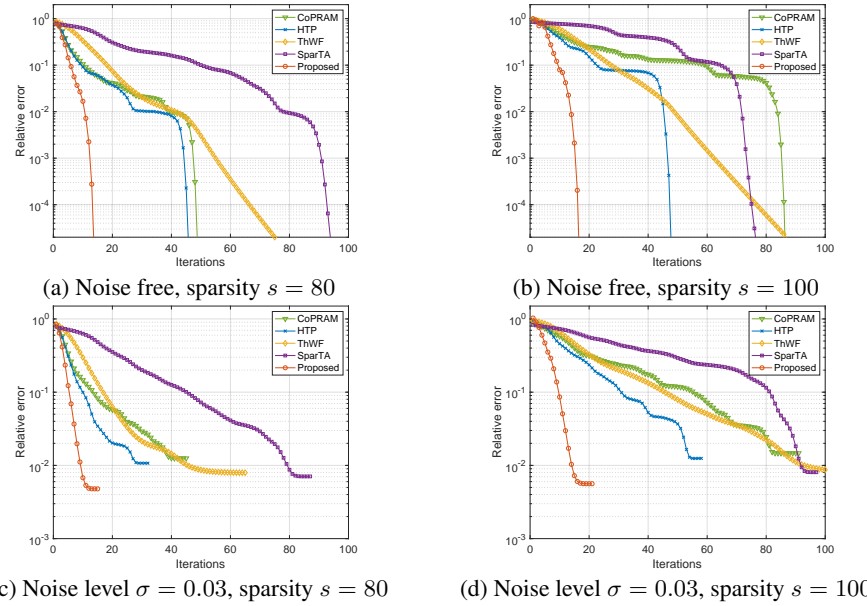

(a) Noise free, sparsity $s = 80$

(b) Noise free, sparsity $s = 100$

(c) Noise level $\sigma = 0.03$, sparsity $s = 80$

(d) Noise level $\sigma = 0.03$, sparsity $s = 100$

Figure 1: Relative error versus iterations for various algorithms, with fixed signal dimension $n = 5000$ and sample size $m = 3000$. The results represent the average of 100 independent trial runs.

Table 2: Comparison of running times (in seconds) for various algorithms in the recovery of signals with sparsity levels of 80 and 100 for both noise-free and noisy scenarios.

| Methods | | ThWF | SPARTA | CoPRAM | HTP | Proposed |
|---|---|---|---|---|---|---|
| Noise free ($\sigma = 0$) | Sparsity 80 | 0.3630 | 1.0059 | 0.9762 | 0.0813 | **0.0530** |
| | Sparsity 100 | 0.6262 | 1.2966 | 3.3326 | 0.2212 | **0.1024** |
| Noisy ($\sigma = 0.03$) | Sparsity 80 | 0.2820 | 1.1082 | 1.3426 | 0.1134 | **0.0803** |
| | Sparsity 100 | 0.4039 | 1.6368 | 4.1006 | 0.2213 | **0.1187** |

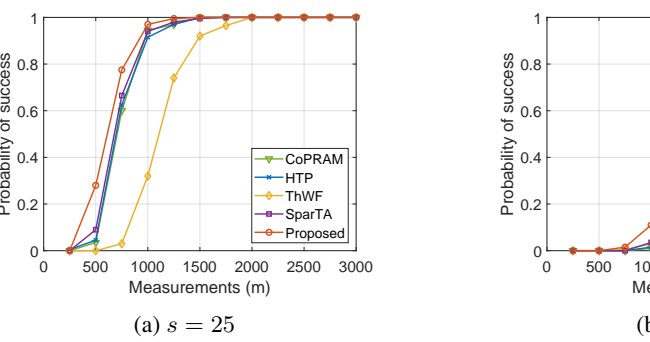

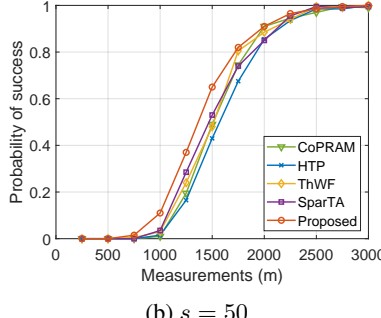

(a) $s = 25$

(b) $s = 50$

Figure 2: Phase transition performance of various algorithms for signals of dimension $n = 3000$ with sparsity levels $s = 25$ and $50$. The results represent the average of 200 independent trial runs. The parameter settings for ThWF, SPARTA, CoPRAM, and HTP in experiments are consistent with those in the studies by Jagatap & Hegde (2019); Cai et al. (2022c).

Figure 2 depicts the phase transitions of different algorithms, with the true signal dimension fixed at $n = 3000$ and sparsity levels set to $s = 25$ and $50$. The phase transition graph is generated by evaluating the successful recovery rate of each algorithm over 200 independent trial runs. Figure 2 shows that the probability of successful recovery for each algorithm transitions from zero to one as the sample size $m$ increases. Furthermore, our algorithm consistently outperforms state-of-the-art methods, achieving a higher successful recovery rate across various measurement counts.

In practical applications, natural signals may not be inherently sparse; however, their wavelet coefficients often exhibit sparsity. Figure 3 illustrates the reconstruction performance of a signal from noisy phaseless measurements, where the true signal, with a dimension of 30,000, exhibits sparsity and contains 208 nonzero entries under the wavelet transform, using 20,000 samples. The sampling matrix $A \in \mathbb{R}^{20,000 \times 30,000}$ is constructed from a random Gaussian matrix and an inverse wavelet transform generated using four levels of Daubechies 1 wavelet. The noise level is set to $\sigma = 0.03$.

To evaluate recovery accuracy, we use the Peak Signal-to-Noise Ratio (PSNR), defined as PSNR = $10 \cdot \log \frac{V^2}{MSE}$, where V represents the maximum fluctuation in the ground truth signal, and MSE denotes the mean squared error of the reconstruction. A higher PSNR value generally indicates better reconstruction quality. As depicted in Figure 3, our proposed algorithm outperforms state-of-the-art methods in terms of both reconstruction time and PSNR. It achieves a higher PSNR while requiring considerably less time for reconstruction. In the experiments, the sparsity level is assumed to be unknown, and the hard thresholding sparsity level is set to 300 for various algorithms.

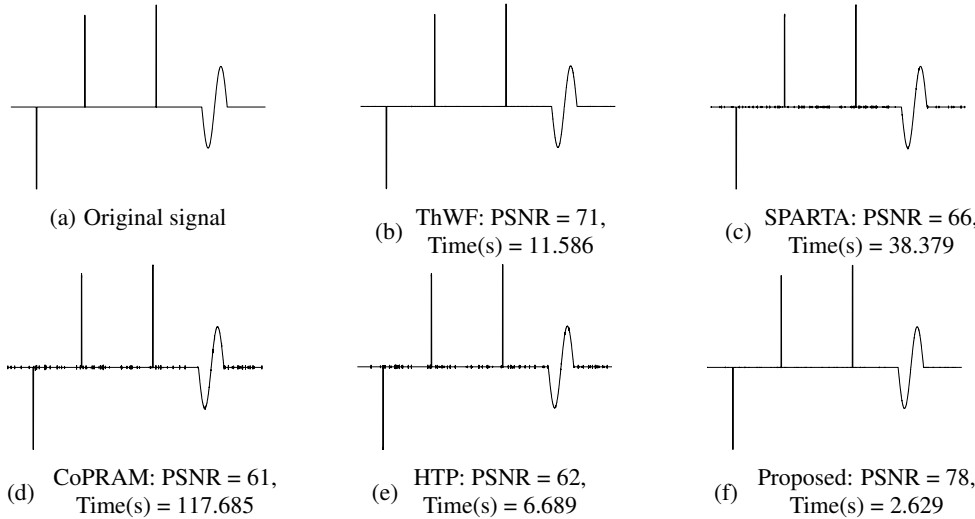

Figure 3: Reconstruction of the signal with a dimension of 30,000 from noisy phaseless measurements by various algorithms. Time(s) is the running time in seconds.

## 5 CONCLUSIONS AND DISCUSSIONS

In this paper, we have introduced an efficient second-order algorithm for sparse phase retrieval. Our algorithm attains a non-asymptotic quadratic convergence rate while maintaining the same per-iteration computational complexity as popular first-order methods, which exhibit linear convergence limitations. Empirical results have demonstrated a significant improvement in the convergence rate of our algorithm. Furthermore, experiments have revealed that our algorithm excels in attaining a higher success rate for exact signal recovery.

Finally, we discuss the limitations of our paper, which also serve as potential avenues for future research. Both our algorithm and state-of-the-art methods share the same sample complexity of $\Omega(s^2 \log n)$ for successful recovery; however, our algorithm requires this complexity in both the initialization and refinement stages, while state-of-the-art methods require $\Omega(s^2 \log n)$ for initialization and $\Omega(s \log n/s)$ for refinement. Investigating ways to achieve tighter complexity in our algorithm's refinement stage is a worthwhile pursuit for future studies.

Currently, the initialization methods for sparse phase retrieval exhibit a sub-optimal sample complexity of $\Omega(s^2 \log n)$. A key challenge involves reducing its quadratic dependence on $s$. Recent study (Jagatap & Hegde, 2019) attained a complexity of $\Omega(s \log n)$, closer to the information-theoretic limit, but relied on the strong assumption of the power law decay for sparse signals. Developing an initialization method that offers optimal sample complexity for a broader range of sparse signals is an engaging direction for future research.

ACKNOWLEDGMENTS

This work was supported by the Hong Kong Research Grant Council GRFs 16310620, 16306821, and 16307023, and Hong Kong Innovation and Technology Fund MHP/009/20, and the Project of Hetao Shenzhen-Hong Kong Science and Technology Innovation Cooperation Zone under Grant HZQB-KCZYB-2020083. We would also like to thank the anonymous reviewers for their valuable feedback on the manuscript.

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

Additional experimental results are provided in Appendix A, while the proofs for Theorems 3.1 and 3.2 can be found in Appendix B.

# A  ADDITIONAL EXPERIMENTS

In this section, we conduct a series of experiments to evaluate the scalability of the proposed algorithm, its phase transition characteristics during signal recovery, its resilience when confronted with various levels of input sparsity, and its robustness in the face of noisy measurements.

## A.1  SCALABILITY ACROSS VARYING DIMENSIONS

We first investigate the scalability of our algorithm in terms of varying dimensions. In particular, we extend the range of signal dimensions from 10000 to 50000 and adjust the sample size ratio $(m/n)$ from 0.3 to 0.7. We define successful recovery as the termination condition for the algorithm, specifically when the iterate fulfills: $\frac{\text{dist}(x^k, x^\natural)}{\|x^\natural\|} < 10^{-3}$. Table 3 offers an in-depth view of the efficiency and scalability of our proposed algorithm.

Table 3: Running time comparison (in seconds) for the proposed algorithm while recovering signals with dimensions ranging from 10000 to 50000 and sample size ratios $(m/n)$ from 0.3 to 0.7. The underlying signal has a sparsity level of 100. The reported results represent the average of 200 independent trial runs.

| Dimension $n$ $(10^5)$ | | 1 | 1.5 | 2 | 2.5 | 3 | 3.5 | 4 | 4.5 | 5 |
|---|---|---|---|---|---|---|---|---|---|---|
| Sample size ratio $m/n$ | 0.3 | 0.212 | 0.420 | 0.458 | 0.675 | 0.987 | 1.214 | 1.472 | 1.854 | 2.147 |
| | 0.5 | 0.242 | 0.497 | 0.585 | 0.912 | 1.382 | 1.788 | 2.237 | 2.709 | 3.022 |
| | 0.7 | 0.278 | 0.545 | 0.833 | 1.179 | 1.783 | 2.224 | 2.789 | 3.581 | 4.252 |

## A.2  PHASE TRANSITIONS FOR BLOCK-SPARSE SIGNALS

We present the phase transitions of various algorithms in Figure 4, focusing on block-sparse signals with a fixed dimension of $n = 3000$ and sparsity levels set to $s = 20$ and 30. The signal generation process aligns with the experiments depicted in Figure 2 of the study by Jagatap & Hegde (2019). We generated the phase transition graph by assessing the success rate of each algorithm's recovery across 200 independent trial runs. As shown in Figure 4, our algorithm always achieves a higher successful recovery rate than the state-of-the-art methods across various measurement counts.

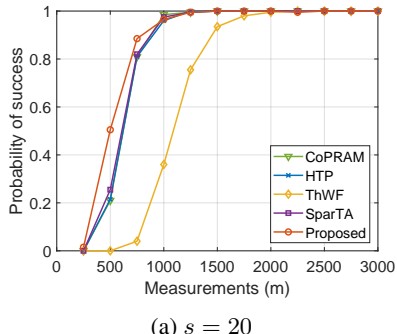
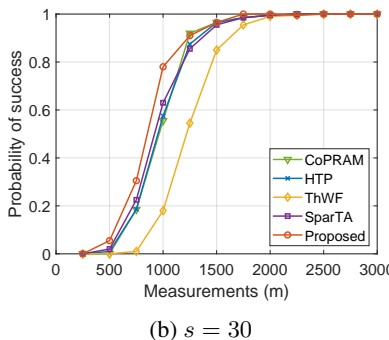

(a) $s = 20$  (b) $s = 30$

Figure 4: Phase transition comparison of various algorithms applied to block-sparse signals with a dimension of $n = 3000$ and sparsity levels $s = 20$ and 30. The signal generation process aligns with the experiments depicted in Figure 2 of the study by Jagatap & Hegde (2019). The parameter settings for ThWF, SPARTA, CoPRAM, and HTP are consistent with those used in the studies by Jagatap & Hegde (2019); Cai et al. (2022c). The results reflect the average of 200 independent trial runs. A recovery is considered successful if the relative error was less than $10^{-3}$.

## A.3 PHASE TRANSITIONS ACROSS VARYING SPARSITY LEVELS

Figure 5 presents the success rate of different algorithms as a function of varying sparsity levels $s$ and the number of measurements $m$. With a fixed signal dimension of $n = 3000$, we vary the signal sparsity $s$ from 6 to 120 and the number of measurements $m$ from 150 to 3000. A signal recovery is considered successful if the relative error $\frac{\text{dist}(\hat{\boldsymbol{x}}, \boldsymbol{x}^\natural)}{\|\boldsymbol{x}^\natural\|} < 10^{-3}$. The gray level of a block represents the success rate: black corresponds to a 0% successful recovery, white to a 100% successful recovery, and gray to a rate between 0% and 100%. As Figure 5 demonstrates, our algorithm outperforms the state-of-the-art methods at higher values of $s$. For lower values of $s$, our algorithm achieves slightly better results compared to ThWF, similar to the performances of SPARTA, CoPRAM, and HTP.

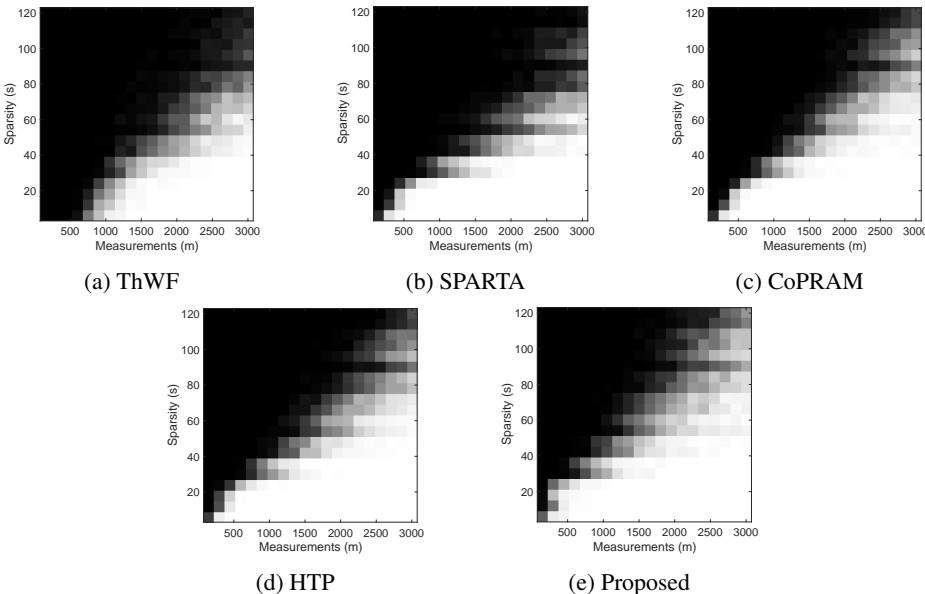

Figure 5: Comparing phase transitions among various algorithms for a signal dimension of $n = 3000$, across different sparsity levels and numbers of measurements. The successful recovery rates are indicated by varying grey levels in the corresponding block. Black signifies a 0% successful recovery rate, white indicates 100%, and grey represents values between 0% and 100%. A signal recovery is considered successful if its relative error is less than $10^{-3}$.

## A.4 PERFORMANCE COMPARISON WITH UNKNOWN SPARSITY

In Table 4, we consider scenarios with unknown sparsity. We input various sparsity levels into each algorithm and compare the success rates of various algorithms in recovering the signal. In these experiments, the underlying signal has a sparsity of 30, a signal dimension of 3000, and the number of measurements is 2000. We excluded ThWF from the comparison because it does not require input sparsity. As demonstrated in Table 4, our proposed algorithm exhibits significant robustness to changes in input sparsity levels.

Table 4: Comparison of success rates of various algorithms with unknown signal sparsity. The underlying signal has a sparsity of 30, a dimension of 3000, and a total of 2000 samples.

| Input sparsity | 10 | 20 | 30 | 50 | 70 | 100 | 150 | 200 | 250 | 300 |
|---|---|---|---|---|---|---|---|---|---|---|
| CoPRAM | 0 | 0 | 1 | 1 | 1 | 1 | 0.75 | 0.09 | 0 | 0 |
| HTP | 0 | 0 | 1 | 1 | 1 | 1 | 0.71 | 0.22 | 0.02 | 0.01 |
| SPARTA | 0 | 0 | 1 | 1 | 1 | 0.09 | 0 | 0 | 0 | 0 |
| Proposed | 0 | 0 | 1 | 1 | 1 | 1 | 0.93 | 0.85 | 0.76 | 0.66 |

## A.5 NOISE ROBUSTNESS

We investigate the impact of the noise level of measurements on the recovery error of our proposed algorithm. Noise levels are represented by signal-to-noise ratios (SNR), *i.e.*, $\|x^\natural\|/\sigma$, where $x^\natural$ is the ground truth signal and $\sigma$ is a parameter determining the standard deviation of Gaussian noise, as defined in (12). We set the dimension of the ground truth signal to $n = 2000$, the sparsity level to $s = 20$, and the number of measurements to $m = 1500$.

Figure 6 depicts the relative error of the proposed algorithm as a function of signal-to-noise ratios (SNR) in dB. We observe a nearly linear decrease in the relative error as the SNR increases, implying that its recovery error can be controlled by a multiple of the measurement noise level. This result demonstrates the robustness of our algorithm against noise in measurements. Additionally, the small error bars shown in Figure 6 emphasize the low variability of recovery errors of our algorithm.

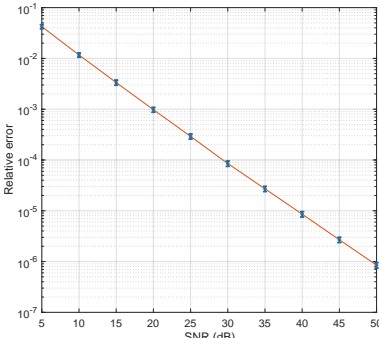

Figure 6: Robustness of the proposed algorithm against additive Gaussian noise. The $y$-axis represents the relative error of the proposed algorithm, while the $x$-axis corresponds to the signal-to-noise ratios (SNR) of the measurements. The results are averaged over 100 independent trial runs, with error bars indicating the standard deviation of the recovery error. We set $n = 2000$, $m = 1500$, and $s = 20$.

## B   PROOFS

We provide proofs for Theorem 3.1, the recovery guarantee for noise-free measurements, and Theorem 3.2, the recovery guarantee for noisy measurements. Technical lemmas are presented in Appendix B.1, which serve as the foundation for proving Theorems 3.1 and 3.2, and their proofs are available in Appendix B.2. Subsequently, the proofs of Theorems 3.1 and 3.2 can be found in Appendix B.3.

For a more concise representation, we arrange the sampling vectors and observations as follows:

$$\boldsymbol{A} := [\boldsymbol{a}_1\ \boldsymbol{a}_2\ \cdots\ \boldsymbol{a}_m]^T, \quad \boldsymbol{y} := [y_1\ y_2\ \cdots\ y_m]^T, \quad \boldsymbol{z} := [z_1\ z_2\ \cdots\ z_m]^T, \tag{13}$$

where $z_i = \sqrt{y_i}$, $i = 1, \ldots, m$. We denote $\boldsymbol{a}_{i,\mathcal{S}}$ as the row vector that represents the $i$-th row of $\boldsymbol{A}$, retaining only the entries indexed by $\mathcal{S}$.

### B.1   TECHNICAL LEMMAS

The following lemma serves as an extension of Lemma 7.4 found in (Candes et al., 2015).

**Lemma B.1.** *For any $s$-sparse vector $x^\natural \in \mathbb{R}^n$ with support $\mathcal{S}^\natural = \mathrm{supp}(x^\natural)$, let $\{a_i\}_{i=1}^m$ be identically and independently distributed as $\mathcal{N}(\boldsymbol{0}, \boldsymbol{I}_n)$ and define the matrix $\boldsymbol{A}$ as in (13). For any subset $\mathcal{S} \subseteq [n]$ such that $\mathrm{supp}(x^\natural) \subseteq \mathcal{S}$ and $|\mathcal{S}| \leq r$ for some integer $r \leq n$. With probability at least $1 - m^{-4} - c_a\delta^{-2}m^{-1} - c_b \exp\left(-c_c\delta^2 m/\log m\right)$, we have*

$$\left\|\frac{1}{m}\sum_{i=1}^m |\boldsymbol{a}_{i,\mathcal{S}}^T x^\natural|^2 \boldsymbol{a}_{i,\mathcal{S}}\boldsymbol{a}_{i,\mathcal{S}}^T - \left(\|x^\natural\|^2(\boldsymbol{I}_n)_{\mathcal{S},\mathcal{S}} + 2x_{\mathcal{S}}^\natural(x_{\mathcal{S}}^\natural)^T\right)\right\| \leq \delta\|x^\natural\|^2 \tag{14}$$

*provided $m \geq C(\delta)r \log(n/r)$. Here $C(\delta)$ is a constant depending on $\delta$, $c_a, c_b$ and $c_c$ are positive absolute constants, and $\boldsymbol{a}_{i,\mathcal{S}}$ represents the $i$-th row of $\boldsymbol{A}$, retaining only the entries indexed by $\mathcal{S}$.*

The following lemma is derived through the application of Hölder's inequality and Lemma B.9:

**Lemma B.2.** *Given two vectors $x, z \in \mathbb{R}^n$ each with sparsity no larger than $s$ and two subsets $\mathcal{S}, \mathcal{T} \subseteq [n]$. Define the subset $\mathcal{R} := \mathcal{S} \cup \mathcal{T} \cup \mathrm{supp}(x) \cup \mathrm{supp}(z)$. Under the event (20) with support $\mathcal{R}$, there holds*

$$\left\| \nabla^2_{\mathcal{S},\mathcal{T}} f_I(x) - \nabla^2_{\mathcal{S},\mathcal{T}} f_I(z) \right\| \leq \frac{3}{m} \left( (3m)^{1/4} + (|\mathcal{R}|)^{1/2} + 2\sqrt{\log m} \right)^4 \|x + z\| \|x - z\|. \quad (15)$$

The following inequalities in Lemma B.3 can be derived using Lemmas B.1, B.2, B.11, and B.12.

**Lemma B.3.** *Let $x^\natural \in \mathbb{R}^n$ be any signal with sparsity $\|x^\natural\|_0 \leq s$ and support $\mathcal{S}^\natural$. Let $\{a_i\}_{i=1}^m$ be random Gaussian vectors identically and independently distributed as $\mathcal{N}(0, I_n)$. Define $A$, $y$ and $f_I(x)$ as in (13) and (4) respectively. Given two subsets $\mathcal{S}, \mathcal{T} \subseteq [n]$ satisfying $|\mathcal{S}| \leq s$ and $|\mathcal{T}| \leq s$. Then if $m \geq 30s^2$, for any $s$-sparse vector $x \in \mathbb{R}^n$ with $\mathrm{supp}(x) \subseteq \mathcal{T}$ and obeying $\mathrm{dist}(x, x^\natural) \leq \gamma \|x^\natural\|$ with $0 < \gamma \leq 0.1$, under events (20) and (14), the following inequalities hold:*

*(i)*
$$l_1 \|u\| \leq \left\| \nabla^2_{\mathcal{S},\mathcal{S}} f_I(x) u \right\| \leq l_2 \|u\|, \quad \forall\, u \in \mathbb{R}^{|\mathcal{S}|}, \quad (16)$$

*where $l_1 := (2 - 2\delta - 10\gamma(2 + \gamma)) \|x^\natural\|^2$ and $l_2 := (6 + 2\delta + 10\gamma(2 + \gamma)) \|x^\natural\|^2$.*

*(ii)*
$$\left\| \nabla^2_{\mathcal{S},\mathcal{S}^\natural \setminus \mathcal{S}} f_I(x) \right\| \leq l_3 := (2 + 2\delta + 10\gamma(2 + \gamma)) \|x^\natural\|^2. \quad (17)$$

The next lemma is adapted from Lemma 3 in (Cai et al., 2022c).

**Lemma B.4.** *Let $\{x^k\}_{k \geq 1}$ be the sequence generated by the Algorithm 1. Let $z^k := z \odot \mathrm{sgn}(Ax^k)$. Assume $\|x^k - x^\natural\| \leq \gamma \|x^\natural\|$. Then under the event (18) with $r = s, 2s$ and the event (19), it holds that*

$$\frac{1}{m} \left\| A^T_{\mathcal{S}_{k+1}} (z^k - Ax^\natural) \right\| \leq \sqrt{C_\gamma (1 + \delta_s)} \|x^k - x^\natural\|,$$

*where $C_\gamma = \frac{4}{(1-\gamma^2)} (\epsilon_0 + \gamma \sqrt{\frac{21}{20}})^2$ with $\epsilon_0 = 10^{-3}$, and $\mathcal{S}_{k+1} = \mathrm{supp}(\mathcal{H}_s(x^k - \eta \nabla f_A(x^k)))$.*

The following lemma provides an upper bound on the estimation error for the vector obtained after one iteration of IHT, as described in (Blumensath & Davies, 2009). To make this paper self-contained, we include the details of the proof for the reader's convenience.

**Lemma B.5.** *Given an $s$-sparse estimate $x^k$ satisfying $\|x^k - x^\natural\| \leq \gamma \|x^\natural\|$. Define the vector obtained by one iteration of IHT with stepsize $\eta$ to be*

$$u^k := \mathcal{H}_s \left( x^k - \eta \nabla f_A (x^k) \right).$$

*Under the RIP event (18), it holds that*

$$\left\| u^k - x^\natural \right\| \leq \zeta \left\| x^k - x^\natural \right\|,$$

*where $\zeta = 2 \left( \sqrt{2} \max\{\eta \delta_{3s}, 1 - \eta(1 - \delta_{2s})\} + \eta \sqrt{C_\gamma (1 + \delta_{2s})} \right)$.*

The following lemma asserts that, given a sufficiently large $m$, the normalized random Gaussian matrix $A$ obeys the restricted isometry property (RIP) with overwhelming probability. This conclusion is well-established in compressive sensing literature (Candes & Tao, 2005; Foucart & Rauhut, 2013).

**Lemma B.6.** *(Foucart & Rauhut, 2013, Theorem 9.27) Let $A$ be defined as in (13) with each vector $a_i$ distributed as the random Gaussian vector $a \sim \mathcal{N}(0, I_n)$ independently for $i = 1, 2, \ldots, m$. There exists some universal positive constants $C_1', C_2'$ such that for any positive integer $r \leq n$ and any $\delta_r \in (0, 1)$, if $m \geq C_1' \delta_r^{-2} r \log(n/r)$, then $\frac{1}{\sqrt{m}} A$ satisfies $r$-RIP with constant $\delta_r$, namely*

$$(1 - \delta_r) \|x\|^2 \leq \left\| \frac{1}{\sqrt{m}} Ax \right\|^2 \leq (1 + \delta_r) \|x\|^2, \quad \forall x \in \mathbb{R}^n \text{with} \|x\|_0 \leq r, \quad (18)$$

*with probability at least $1 - e^{-C_2' m}$.*

The results presented below have been previously established in (Cai et al., 2016).

**Lemma B.7.** *(Cai et al., 2016, Lemma A.6) On an event with probability at least $1 - m^{-1}$, we have*

$$\left\| \frac{1}{m} \sum_{i=1}^{m} |a_{i,\mathcal{S}^\natural}^T x^\natural|^2 a_{i,\mathcal{S}^\natural} a_{i,\mathcal{S}^\natural}^T - \left( \|x^\natural\|^2 (I_n)_{\mathcal{S}^\natural, \mathcal{S}^\natural} + 2 x_{\mathcal{S}^\natural}^\natural (x_{\mathcal{S}^\natural}^\natural)^T \right) \right\| \leq \delta \|x^\natural\|^2,$$

*provided $m \geq C(\delta) s \log s$. Here $C(\delta)$ is a constant depending on $\delta$.*

The subsequent lemma, a direct outcome from (Soltanolkotabi, 2019), plays a crucial role in bounding the term $\left\| A_{\mathcal{T}_{k+1}}^T (z^k - A x^\natural) \right\|$.

**Lemma B.8.** *(Soltanolkotabi, 2019, Lemma 25) Let $\{a_i\}_{i=1}^m$ be i.i.d. Gaussian random vectors with mean $\mathbf{0}$ and variance matrix $I$. Let $\gamma$ be any constant in $(0, \frac{1}{8}]$. Fixing any $\epsilon_0 > 0$, then for any $s$-sparse vector $x$ satisfying $\mathrm{dist}(x, x^\natural) \leq \lambda_0 \|x^\natural\|$, with probability at least $1 - e^{-C_6' m}$ there holds that*

$$\frac{1}{m} \sum_{i=1}^{m} |a_i^T x^\natural|^2 \cdot \mathbf{1}_{\{(a_i^T x)(a_i^T x^\natural) \leq 0\}} \leq \frac{1}{(1 - \lambda_0)^2} \left( \epsilon_0 + \lambda_0 \sqrt{\frac{21}{20}} \right)^2 \|x - x^\natural\|^2, \qquad (19)$$

*provided $m \geq C_5' s \log(n/s)$. Here $C_5'$ and $C_6'$ are some universal positive constants.*

The next lemma, derived from (Cai et al., 2016), asserts that the induced norm of the submatrix of random Gaussian matrix $A$ is bounded by its size.

**Lemma B.9.** *(Cai et al., 2016, Lemma A.5) Let $\{a_i\}_{i=1}^m$ be identically and independently distributed as $\mathcal{N}(\mathbf{0}, I_n)$ and define the matrix $A$ as in (13). Given a support $\mathcal{S} \subseteq [n]$ with cardinality $s$, with probability at least $1 - 4m^{-2}$, there holds*

$$\|A_{\mathcal{S}}\|_{2 \to 4} \leq (3m)^{1/4} + s^{1/2} + 2\sqrt{\log m}, \qquad (20)$$

*where $A_{\mathcal{S}}$ represents the sub-matrix of $A$, retaining only the columns indexed by $\mathcal{S}$.*

The subsequent result concerning the spectral norm of submatrices of $\frac{1}{\sqrt{m}} A$ can be deduced by employing the restricted isometry property of matrix $\frac{1}{\sqrt{m}} A$.

**Lemma B.10.** *(Needell & Tropp, 2009, Proposition 3.1) Suppose the matrix $A$ satisfies the inequality (18) for values of $r$ specified as $s$ and $s'$. Then for any disjoint subsets $\mathcal{S}$ and $\mathcal{T}$ of $\{1, 2, \cdots, m\}$ satisfying $|\mathcal{S}| \leq s$ and $|\mathcal{T}| \leq s'$, the following inequalities hold:*

$$\left\| \frac{1}{\sqrt{m}} A_{\mathcal{S}}^T \right\| \leq \sqrt{1 + \delta_s}, \qquad (21)$$

$$\left\| \frac{1}{m} A_{\mathcal{S}}^T A_{\mathcal{T}} \right\| \leq \delta_{s+s'}, \qquad (22)$$

$$1 - \delta_s \leq \left\| \frac{1}{m} A_{\mathcal{S}}^T A_{\mathcal{S}} \right\| \leq 1 + \delta_s, . \qquad (23)$$

The following lemma provides the expectation of the sub-Hessian of the intensity-based loss $f_I$ at $x^\natural$. As this result can be easily derived through basic calculations, we will not delve into the details here.

**Lemma B.11.** *For any subset $\mathcal{S} \subseteq [n]$ such that $\mathrm{supp}(x^\natural) \subseteq \mathcal{S}$, the expectation of $\nabla_{\mathcal{S}, \mathcal{S}}^2 f_I(x^\natural)$ is*

$$\mathbb{E} \left[ \nabla_{\mathcal{S}, \mathcal{S}}^2 f_I(x^\natural) \right] = 2 \left( \|x^\natural\|^2 (I_n)_{\mathcal{S}, \mathcal{S}} + 2 x_{\mathcal{S}}^\natural (x_{\mathcal{S}}^\natural)^T \right),$$

*and it has one eigenvalue of $6\|x^\natural\|^2$ and all other eigenvalues are $2\|x^\natural\|^2$.*

The next lemma is the so-called Weyl Theorem, which is a classical linear algebra result.

**Lemma B.12.** *Suppose $M$, $N \in \mathbb{R}^{n \times n}$ are two symmetric matrices. The eigenvalues of $M$ are denoted as $\lambda_1 \geq \lambda_2 \geq \cdots \lambda_n$ and the eigenvalues of $N$ are denoted as $\mu_1 \geq \mu_2 \geq \cdots \mu_n$. Then we have*

$$|\mu_i - \lambda_i| \leq \|M - N\|_2, \quad \forall i = 1, 2, \cdots, n.$$

**Lemma B.13.** *(Hoeffding-type inequality) Let $X_1, \cdots, X_N$ be independent centered sub-Gaussian random variables, and let $K = \max_{i \in [N]} \|X_i\|_{\psi_2}$, where the sub-Gaussian norm*

$$\|X_i\|_{\psi_2} := \sup_{p \geq 1} p^{-1/2} \left( \mathbb{E}\left[|X|^p\right] \right)^{1/p}.$$

*Then for every $b = [b_1; \cdots ; b_N] \in \mathbb{R}^N$ and every $t \geq 0$, we have*

$$\mathbb{P}\left\{ \left| \sum_{k=1}^{N} b_k X_k \right| \geq t \right\} \leq e \exp\left( -\frac{ct^2}{K^2 \|b\|^2} \right).$$

*Here $c$ is a universal constant.*

**Lemma B.14.** *(Bernstein-type inequality) Let $X_1, \cdots, X_N$ be independent centered sub-exponential random variables, and let $K = \max_i \|X_i\|_{\psi_1}$, where the sub-exponential norm*

$$\|X_i\|_{\psi_1} := \sup_{p \geq 1} p^{-1} \left( \mathbb{E}\left[|X|^p\right] \right)^{1/p}.$$

*Then for every $b = [b_1; \cdots ; b_N] \in \mathbb{R}^N$ and every $t \geq 0$, we have*

$$\mathbb{P}\left\{ \left| \sum_{k=1}^{N} b_k X_k \right| \geq t \right\} \leq 2 \exp\left( -c \min\left( \frac{t^2}{K^2 \|b\|^2}, \frac{t}{K \|b\|_\infty} \right) \right).$$

*Here $c$ is a universal constant.*

**Lemma B.15.** *(Bernstein's inequality for bounded distributions) Let $X_1, \cdots, X_N$ be independent mean zero random variables, such that $|X_i| \leq K$ for all $i$. Then, for every $t \geq 0$, we have*

$$\mathbb{P}\left\{ \left| \sum_{i=1}^{N} X_i \right| \geq t \right\} \leq 2 \exp\left( -\frac{t^2/2}{\sigma^2 + Kt/3} \right).$$

*Here $\sigma^2 = \sum_{i=1}^{N} \mathbb{E} X_i^2$ is the variance of the sum.*

## B.2 PROOFS OF TECHNICAL LEMMAS

In this subsection, we provide proofs for the technical lemmas: Lemmas B.1, B.2, B.3, B.4, and B.5.

### B.2.1 PROOF OF LEMMA B.1

*Proof of Lemma B.1.* If $|\mathcal{S}| = |\mathcal{S}^\natural|$, then $\mathcal{S} = \mathcal{S}^\natural$ and the result is established by Lemma B.7. We now focus on the case where $\mathcal{T} := \mathcal{S} \backslash \mathcal{S}^\natural \neq \emptyset$. We first define two matrices as follows:

$$G = \begin{bmatrix} a_{i,\mathcal{S}^\natural} a_{i,\mathcal{S}^\natural}^T & a_{i,\mathcal{S}^\natural} a_{i,\mathcal{T}}^T \\ a_{i,\mathcal{T}} a_{i,\mathcal{S}^\natural}^T & a_{i,\mathcal{T}} a_{i,\mathcal{T}}^T \end{bmatrix}, \quad H = \begin{bmatrix} \|x^\natural\|^2 (I_n)_{\mathcal{S}^\natural,\mathcal{S}^\natural} + 2 x_{\mathcal{S}^\natural}^\natural (x_{\mathcal{S}^\natural}^\natural)^T & 0 \\ 0 & \|x^\natural\|^2 (I_n)_{\mathcal{T},\mathcal{T}} \end{bmatrix}.$$

Then we rephrase the term on the left-hand side of (14) as follows:

$$\left\| \frac{1}{m} \sum_{i=1}^{m} |a_{i,\mathcal{S}^\natural}^T x^\natural|^2 G - H \right\| \leq \left\| \frac{1}{m} \sum_{i=1}^{m} |a_{i,\mathcal{S}^\natural}^T x^\natural|^2 a_{i,\mathcal{S}^\natural} a_{i,\mathcal{S}^\natural}^T - \left( \|x^\natural\|^2 (I_n)_{\mathcal{S}^\natural,\mathcal{S}^\natural} + 2 x_{\mathcal{S}^\natural}^\natural (x_{\mathcal{S}^\natural}^\natural)^T \right) \right\|$$

$$+ \left\| \frac{1}{m} \sum_{i=1}^{m} |a_{i,\mathcal{S}^\natural}^T x^\natural|^2 a_{i,\mathcal{S}^\natural} a_{i,\mathcal{T}}^T \right\| + \left\| \frac{1}{m} \sum_{i=1}^{m} |a_{i,\mathcal{S}^\natural}^T x^\natural|^2 a_{i,\mathcal{T}} a_{i,\mathcal{T}}^T - \|x^\natural\|^2 (I_n)_{\mathcal{T},\mathcal{T}} \right\|.$$

The first term can be bounded with overwhelming probability via a direct application of Lemma B.7 as below whenever $m \geq c_1 \delta^{-2} s \log(n/s)$:

$$\left\| \frac{1}{m} \sum_{i=1}^{m} |\boldsymbol{a}_{i,\mathcal{S}^\natural}^T \boldsymbol{x}^\natural|^2 \boldsymbol{a}_{i,\mathcal{S}^\natural} \boldsymbol{a}_{i,\mathcal{S}^\natural}^T - \left( \|\boldsymbol{x}^\natural\|^2 (\boldsymbol{I}_n)_{\mathcal{S}^\natural, \mathcal{S}^\natural} + 2\boldsymbol{x}_{\mathcal{S}^\natural}^\natural (\boldsymbol{x}_{\mathcal{S}^\natural}^\natural)^T \right) \right\| \leq \delta/4.$$

For the other two terms, it is enough to consider $\boldsymbol{x}^\natural = \boldsymbol{e}_1$ because the unitary invariance of the Gaussian measure and rescaling. Then for a prefixed subset $\mathcal{S}$ (thus $\mathcal{T}$ is also fixed), there exists a unit vector $\boldsymbol{u} \in \mathbb{R}^n$ with $\mathrm{supp}(\boldsymbol{u}) \subseteq \mathcal{T}$ such that the second term is equal to

$$\left\| \frac{1}{m} \sum_{i=1}^{m} (a_i(1))^3 \boldsymbol{a}_{i,\mathcal{T}}^T \right\| = \frac{1}{m} \sum_{i=1}^{m} (a_i(1))^3 (\boldsymbol{a}_i^T \boldsymbol{u}).$$

We emphasize that $\boldsymbol{a}_i^T \boldsymbol{u} = \boldsymbol{a}_{i,\mathcal{T}}^T \boldsymbol{u}_\mathcal{T}$ is a random Gaussian variable distributed as $\mathcal{N}(0,1)$ for all $\boldsymbol{u} \in \mathbb{R}^n, \|\boldsymbol{u}\| = 1, \mathrm{supp}(\boldsymbol{u}) \subseteq \mathcal{T}$ and independent of $a_i(1)$ for all $i \in [m]$. So for one realization of $\{a_i(1)\}$, Lemma B.13 (Hoeffding-type inequality) implies

$$\mathbb{P}\left\{ \left| \frac{1}{m} \sum_{i=1}^{m} (a_i(1))^3 \boldsymbol{a}_i^T \boldsymbol{u} \right| > t \right\} \leq e \exp\left( -\frac{c_2 m^2 t^2}{\sum_{i=1}^{m} |a_i(1)|^6} \right),$$

for any $t > 0$. Define the set $\mathbb{W}_r^n$ to be a collection of all the index set in $[n]$ with cardinality no larger than $r$, i.e. $\mathbb{W}_r^n := \{\mathcal{S} \subseteq [n] : |\mathcal{S}| \leq r\}$. Note that the cardinality of $\mathbb{W}_r^n$ can be bounded by $\sum_{j=1}^{r} \binom{n}{j} \leq \left(\frac{en}{r}\right)^r$. Taking $t = \delta/8$, together with a union bound on the set $\mathbb{W}_r^n$, we obtain for any subset $\mathcal{T} = \mathcal{S} \backslash \mathcal{S}^\natural$,

$$\mathbb{P}\left\{ \left\| \frac{1}{m} \sum_{i=1}^{m} (a_i(1))^3 \boldsymbol{a}_{i,\mathcal{T}}^T \right\| > \delta/4 \right\} \leq e \exp\left( -\frac{c_2 m^2 \delta^2}{64 \sum_{i=1}^{m} |a_i(1)|^6} + 6r \log(n/r) \right).$$

Now with an application of Chebyshev's inequality, we have $\sum_{i=1}^{m} |a_i(1)|^6 \leq 20m$ with probability at least $1 - c_3 m^{-1}$. Substituting this into the above, we conclude that for any subset $\mathcal{S} \subseteq \mathbb{W}_r^n$, whenever $m \geq c_4 \delta^{-2} r \log(n/r)$ for some sufficiently large $c_4$,

$$\left\| \frac{1}{m} \sum_{i=1}^{m} (a_i(1))^3 \boldsymbol{a}_{i,\mathcal{T}}^T \right\| \leq \delta/4,$$

with probability at least $1 - c_3 m^{-1} - e \exp(-c_5 \delta^2 m)$.

Assuming $\boldsymbol{x}^\natural = \boldsymbol{e}_1$ and fixing a subset $\mathcal{S} \subseteq [n]$ (thus $\mathcal{T}$ is also fixed), we can obtain:

$$\left\| \frac{1}{m} \sum_{i=1}^{m} |a_i(1)|^2 \boldsymbol{a}_{i,\mathcal{T}} \boldsymbol{a}_{i,\mathcal{T}}^T - (\boldsymbol{I}_n)_{\mathcal{T},\mathcal{T}} \right\| \leq \left\| \frac{1}{m} \sum_{i=1}^{m} |a_i(1)|^2 \left( \boldsymbol{a}_{i,\mathcal{T}} \boldsymbol{a}_{i,\mathcal{T}}^T - \boldsymbol{I}_{|\mathcal{T}|} \right) \right\| + \left| \frac{1}{m} \sum_{i=1}^{m} |a_i(1)|^2 - 1 \right|,$$

for which there exists a unit vector $\boldsymbol{u} \in \mathbb{R}^n$ with $\mathrm{supp}(\boldsymbol{u}) \subseteq \mathcal{T}$ such that

$$\left\| \frac{1}{m} \sum_{i=1}^{m} |a_i(1)|^2 \left( \boldsymbol{a}_{i,\mathcal{T}} \boldsymbol{a}_{i,\mathcal{T}}^T - \boldsymbol{I}_{|\mathcal{T}|} \right) \right\| = \frac{1}{m} \sum_{i=1}^{m} |a_i(1)|^2 \left| \left( \boldsymbol{a}_{i,\mathcal{T}}^T \boldsymbol{u}_\mathcal{T} \right)^2 - 1 \right|.$$

Note that $\{a_i(1)\}$ is independent of $\{\boldsymbol{a}_{i,\mathcal{T}}\}$, an application of Bernstein's inequality implies

$$\mathbb{P}\left\{ \frac{1}{m} \sum_{i=1}^{m} |a_i(1)|^2 \left| \left( \boldsymbol{a}_{i,\mathcal{T}}^T \boldsymbol{u}_\mathcal{T} \right)^2 - 1 \right| > t \right\} \leq 2 \exp\left( -c_6 \min(d_1, d_2) \right),$$

where $d_1 = \frac{t^2}{c_7^2 \sum_{i=1}^{m} |a_i(1)|^4}$, and $d_2 = \frac{t}{c_7 \max_{i \in [m]} |a_i(1)|^2}$. Taking $t = \delta/8$, together with a union bound on the subset $\mathbb{W}_r^n$, we obtain for any subset $\mathcal{S} \in \mathbb{W}_r^n$,

$$\mathbb{P}\left\{ \left\| \frac{1}{m} \sum_{i=1}^{m} |a_i(1)|^2 \left( \boldsymbol{a}_{i,\mathcal{T}} \boldsymbol{a}_{i,\mathcal{T}}^T - \boldsymbol{I}_{|\mathcal{T}|} \right) \right\| > \delta/4 \right\} \leq 2 \exp\left( -c_6 \min(d_3, d_4) + 6r \log(n/r) \right),$$

where $d_3 = \frac{m^2\delta^2}{64c_7^2 \sum_{i=1}^{m} |a_i(1)|^4}$ and $d_4 = \frac{m\delta}{8c_7 \max_{i\in[m]} |a_i(1)|^2}$. Applying Chebyshev's inequality and the union bound, we obtain:

$$\sum_{i=1}^{m} |a_i(1)|^4 \le 10m \quad \text{and} \quad \max_{i\in[m]} |a_i(1)|^2 \le 10\log m$$

hold with probability at least $1 - c_8 m^{-1} - m^{-4}$. To conclude, for any subset $\mathcal{S} \in \mathbb{W}_r^n$, when $m \ge c_9 \delta^{-2} r \log(n/r)$ for some sufficiently large constant $c_9$,

$$\left\| \frac{1}{m} \sum_{i=1}^{m} |a_i(1)|^2 \left( \boldsymbol{a}_{i,\mathcal{T}} \boldsymbol{a}_{i,\mathcal{T}}^T - \boldsymbol{I}_{|\mathcal{T}|} \right) \right\| \le \delta/4$$

with probability at least $1 - c_8 m^{-1} - m^{-4} - 2\exp\left(-c_{10}\delta^2 m/\log m\right)$. Finally, an application of Chebyshev's inequality implies

$$\left| \frac{1}{m} \sum_{i=1}^{m} |a_i(1)|^2 - 1 \right| \le \delta/4$$

with probability at least $1 - c_{11}\delta^{-2}m^{-1}$. The proof is finished by combining the above bounds and probabilities. $\qquad\square$

### B.2.2 PROOF OF LEMMA B.2

*Proof of Lemma B.2.* A simple calculation yields

$$\nabla_{\mathcal{S},\mathcal{T}}^2 f_I(\boldsymbol{x}) - \nabla_{\mathcal{S},\mathcal{T}}^2 f_I(\boldsymbol{z}) = \frac{3}{m} \boldsymbol{A}_{\mathcal{S}}^T D(|\boldsymbol{A}\boldsymbol{x}|^2 - |\boldsymbol{A}\boldsymbol{z}|^2) \boldsymbol{A}_{\mathcal{T}}.$$

There exist unit vectors $\boldsymbol{u}, \boldsymbol{v} \in \mathbb{R}^n$ with support $\text{supp}(\boldsymbol{u}) \subseteq \mathcal{S}$ and $\text{supp}(\boldsymbol{v}) \subseteq \mathcal{T}$ such that

$$
\begin{aligned}
&\left\| \nabla_{\mathcal{S},\mathcal{T}}^2 f_I(\boldsymbol{x}) - \nabla_{\mathcal{S},\mathcal{T}}^2 f_I(\boldsymbol{z}) \right\| \\
=& \frac{3}{m} \left\| \boldsymbol{A}_{\mathcal{S}}^T D(|\boldsymbol{A}\boldsymbol{x}|^2 - |\boldsymbol{A}\boldsymbol{z}|^2) \boldsymbol{A}_{\mathcal{T}} \right\| \\
=& \frac{3}{m} \left| \sum_{i=1}^{m} \left( |\boldsymbol{a}_i^T \boldsymbol{x}|^2 - |\boldsymbol{a}_i^T \boldsymbol{z}|^2 \right) (\boldsymbol{a}_{i,\mathcal{S}}^T \boldsymbol{u}_{\mathcal{S}})(\boldsymbol{a}_{i,\mathcal{T}}^T \boldsymbol{v}_{\mathcal{T}}) \right| \\
=& \frac{3}{m} \left| \sum_{i=1}^{m} \left( |\boldsymbol{a}_{i,\mathcal{R}}^T \boldsymbol{x}_{\mathcal{R}}|^2 - |\boldsymbol{a}_{i,\mathcal{R}}^T \boldsymbol{z}_{\mathcal{R}}|^2 \right) (\boldsymbol{a}_{i,\mathcal{R}}^T \boldsymbol{u}_{\mathcal{R}})(\boldsymbol{a}_{i,\mathcal{R}}^T \boldsymbol{v}_{\mathcal{R}}) \right| \\
\le& \frac{3}{m} \sum_{i=1}^{m} \left| \left( \boldsymbol{a}_{i,\mathcal{R}}^T (\boldsymbol{x}_{\mathcal{R}} + \boldsymbol{z}_{\mathcal{R}}) \right) \left( \boldsymbol{a}_{i,\mathcal{R}}^T (\boldsymbol{x}_{\mathcal{R}} - \boldsymbol{z}_{\mathcal{R}}) \right) (\boldsymbol{a}_{i,\mathcal{R}}^T \boldsymbol{u}_{\mathcal{R}})(\boldsymbol{a}_{i,\mathcal{R}}^T \boldsymbol{v}_{\mathcal{R}}) \right| \\
\le& \frac{3}{m} \| \boldsymbol{A}_{\mathcal{R}} \|_{2\to4}^4 \| \boldsymbol{x} + \boldsymbol{z} \| \| \boldsymbol{x} - \boldsymbol{z} \| \\
\le& \frac{3}{m} \left( (3m)^{\frac{1}{4}} + (|\mathcal{R}|)^{1/2} + 2\sqrt{\log m} \right)^4 \| \boldsymbol{x} + \boldsymbol{z} \| \| \boldsymbol{x} - \boldsymbol{z} \|,
\end{aligned}
$$

where the last inequality is implied by Lemma B.9. $\qquad\square$

### B.2.3 PROOF OF LEMMA B.3

*Proof of Lemma B.3.* We begin by proving result (i) in Lemma B.3. Define $\mathcal{R} = \mathcal{T} \cup \mathcal{S} \cup \mathcal{S}^\natural$. Then $|\mathcal{R}| \le 3s$. Applying Lemma B.2 yields that

$$
\begin{aligned}
\left\| \nabla_{\mathcal{R},\mathcal{R}}^2 f_I(\boldsymbol{x}) - \nabla_{\mathcal{R},\mathcal{R}}^2 f_I(\boldsymbol{x}^\natural) \right\| &\le \frac{3}{m} \left( (3m)^{1/4} + (3s)^{1/2} + 2\sqrt{\log m} \right)^4 \| \boldsymbol{x} + \boldsymbol{x}^\natural \| \| \boldsymbol{x} - \boldsymbol{x}^\natural \| \\
&\le 10 \| \boldsymbol{x} + \boldsymbol{x}^\natural \| \| \boldsymbol{x} - \boldsymbol{x}^\natural \|,
\end{aligned}
$$

provided $m \geq 30s^2$. Furthermore, following from Lemmas B.1, B.12, and B.11, and the interlacing inequality, we obtain that:

$$\lambda_{\min}(\nabla^2_{\mathcal{S},\mathcal{S}} f_I(\boldsymbol{x})) \geq \lambda_{\min}\left(\nabla^2_{\mathcal{R},\mathcal{R}} f_I(\boldsymbol{x})\right)$$

$$\geq \lambda_{\min}\left(\nabla^2_{\mathcal{R},\mathcal{R}} f_I\left(\boldsymbol{x}^\natural\right)\right) - \left\|\nabla^2_{\mathcal{R},\mathcal{R}} f_I(\boldsymbol{x}) - \nabla^2_{\mathcal{R}} f_I\left(\boldsymbol{x}^\natural\right)\right\|$$

$$\geq \lambda_{\min}\left(\mathbb{E}\left[\nabla^2_{\mathcal{R},\mathcal{R}} f_I\left(\boldsymbol{x}^\natural\right)\right]\right) - \left\|\nabla^2_{\mathcal{R},\mathcal{R}} f_I\left(\boldsymbol{x}^\natural\right) - \mathbb{E}\left[\nabla^2_{\mathcal{R},\mathcal{R}} f_I\left(\boldsymbol{x}^\natural\right)\right]\right\| - 10\left\|\boldsymbol{x} + \boldsymbol{x}^\natural\right\|\left\|\boldsymbol{x} - \boldsymbol{x}^\natural\right\|$$

$$\geq 2\left\|\boldsymbol{x}^\natural\right\|^2 - 2\delta\left\|\boldsymbol{x}^\natural\right\|^2 - 10\left\|\boldsymbol{x} + \boldsymbol{x}^\natural\right\|\left\|\boldsymbol{x} - \boldsymbol{x}^\natural\right\|$$

$$\geq (2 - 2\delta - 10\gamma(2 + \gamma))\left\|\boldsymbol{x}^\natural\right\|^2,$$

where the last inequality is implied by $\text{dist}(\boldsymbol{x}, \boldsymbol{x}^\natural) \leq \gamma\|\boldsymbol{x}^\natural\|$. Similarly, the upper bound for the largest eigenvalue of $\nabla^2_{\mathcal{S}} f_I(\boldsymbol{x})$ can be bounded as follows:

$$\lambda_{\max}(\nabla^2_{\mathcal{S},\mathcal{S}} f_I(\boldsymbol{x})) \leq \lambda_{\max}\left(\nabla^2_{\mathcal{R},\mathcal{R}} f_I(\boldsymbol{x})\right)$$

$$\leq \lambda_{\max}\left(\nabla^2_{\mathcal{R},\mathcal{R}} f_I\left(\boldsymbol{x}^\natural\right)\right) + \left\|\nabla^2_{\mathcal{R},\mathcal{R}} f_I(\boldsymbol{x}) - \nabla^2_{\mathcal{R},\mathcal{R}} f_I\left(\boldsymbol{x}^\natural\right)\right\|$$

$$\leq \lambda_{\max}\left(\mathbb{E}\left[\nabla^2_{\mathcal{R},\mathcal{R}} f_I\left(\boldsymbol{x}^\natural\right)\right]\right) + \left\|\nabla^2_{\mathcal{R},\mathcal{R}} f_I\left(\boldsymbol{x}^\natural\right) - \mathbb{E}\left[\nabla^2_{\mathcal{R},\mathcal{R}} f_I\left(\boldsymbol{x}^\natural\right)\right]\right\| + 10\left\|\boldsymbol{x} + \boldsymbol{x}^\natural\right\|\left\|\boldsymbol{x} - \boldsymbol{x}^\natural\right\|$$

$$\leq 6\left\|\boldsymbol{x}^\natural\right\|^2 + 2\delta\left\|\boldsymbol{x}^\natural\right\|^2 + 10\left\|\boldsymbol{x} + \boldsymbol{x}^\natural\right\|\left\|\boldsymbol{x} - \boldsymbol{x}^\natural\right\|$$

$$\leq (6 + 2\delta + 10\gamma(2 + \gamma))\left\|\boldsymbol{x}^\natural\right\|^2.$$

Now we turn to proving result (ii) in Lemma B.3. Define $\mathcal{R} = \mathcal{T} \cup \mathcal{S} \cup \mathcal{S}^\natural$. Thus $|\mathcal{R}| \leq 3s$. For the disjoint subsets $\mathcal{S}$ and $\mathcal{S}^\natural \backslash \mathcal{S}$, we consider $\nabla^2_{\mathcal{S},\mathcal{S}^\natural \backslash \mathcal{S}} f_I(\boldsymbol{x})$, which is a submatrix of $\nabla^2_{\mathcal{R},\mathcal{R}} f_I(\boldsymbol{x}) - 4\|\boldsymbol{x}^\natural\|^2\boldsymbol{I}$. We note that the spectral norm of a submatrix never exceeds the norm of the entire matrix. By employing the result from part (i), we can deduce that

$$\left\|\nabla^2_{\mathcal{S},\mathcal{S}^\natural \backslash \mathcal{S}} f_I(\boldsymbol{x})\right\| \leq \left\|\nabla^2_{\mathcal{R},\mathcal{R}} f_I(\boldsymbol{x}) - 4\|\boldsymbol{x}^\natural\|^2\boldsymbol{I}\right\|$$

$$\leq \max\{6 + 2\delta + 10\gamma(2 + \gamma) - 4, 4 - (2 - 2\delta - 10\gamma(2 + \gamma))\} \cdot \|\boldsymbol{x}^\natural\|^2$$

$$= (2 + 2\delta + 10\gamma(2 + \gamma))\|\boldsymbol{x}^\natural\|^2,$$

completing the proof. $\qquad\square$

### B.2.4 PROOF OF LEMMA B.4

*Proof of Lemma B.4.* Define the sets $\{\mathcal{G}_k\}_{k \geq 1}$ as follows:

$$\mathcal{G}_k = \{i \mid \text{sgn}(\boldsymbol{a}_i^T \boldsymbol{x}^k) = \text{sgn}(\boldsymbol{a}_i^T \boldsymbol{x}^\natural), 1 \leq i \leq m\}.$$

With $\boldsymbol{z}^k := \boldsymbol{z} \odot sgn(\boldsymbol{A}\boldsymbol{x}^k)$, where $\boldsymbol{z}$ is defined in (13), we deduce that

$$\left(\frac{1}{\sqrt{m}}\left\|\boldsymbol{z}^k - \boldsymbol{A}\boldsymbol{x}^\natural\right\|\right)^2 = \frac{1}{m}\sum_{i=1}^m \left(\text{sgn}\left(\boldsymbol{a}_i^T \boldsymbol{x}^k\right) - \text{sgn}\left(\boldsymbol{a}_i^T \boldsymbol{x}^\natural\right)\right)^2 \left|\boldsymbol{a}_i^T \boldsymbol{x}^\natural\right|^2$$

$$\leq \frac{4}{m}\sum_{i \in \mathcal{G}_k^c} \left|\boldsymbol{a}_i^T \boldsymbol{x}^\natural\right|^2 \cdot \mathbf{1}_{(\boldsymbol{a}_i^T \boldsymbol{x}^k)(\boldsymbol{a}_i^T \boldsymbol{x}^\natural) \leq 0} \tag{24}$$

$$\leq \underbrace{\frac{4}{(1 - \gamma)^2}\left(\epsilon_0 + \gamma\sqrt{\frac{21}{20}}\right)^2}_{C_\gamma} \left\|\boldsymbol{x}^k - \boldsymbol{x}^\natural\right\|^2,$$

where the first inequality follows from $|\text{sgn}(\boldsymbol{a}_i^T \boldsymbol{x}^k) - \text{sgn}(\boldsymbol{a}_i^T \boldsymbol{x}^\natural)| \leq 2$ and $\text{sgn}(\boldsymbol{a}_i^T \boldsymbol{x}^k) - \text{sgn}(\boldsymbol{a}_i^T \boldsymbol{x}^\natural) = 0$ on $\mathcal{G}_k$, and the second inequality follows from Lemma B.8. Together with (21) in Lemma B.10, (24) leads to

$$\frac{1}{m}\left\|\boldsymbol{A}_{\mathcal{T}_{k+1}}^T(\boldsymbol{z}^k - \boldsymbol{A}\boldsymbol{x}^\natural)\right\| \leq \sqrt{C_\gamma(1 + \delta_s)}\left\|\boldsymbol{x}^k - \boldsymbol{x}^\natural\right\|,$$

completing the proof. $\qquad\square$

### B.2.5 Proof of Lemma B.5

*Proof of Lemma B.5.* Define $\mathcal{S}^{\natural} := \mathrm{supp}\left(\boldsymbol{x}^{\natural}\right)$, $\mathcal{T}_{k+1} := \mathcal{S}_{k+1} \cup \mathcal{S}^{\natural}$, and $\boldsymbol{v}^k := \boldsymbol{x}^k - \eta \nabla f_A\left(\boldsymbol{x}^k\right)$. Since $\boldsymbol{u}^k$ is the best $s$-term approximation of $\boldsymbol{v}^k$, we have

$$\left\|\boldsymbol{u}^k - \boldsymbol{v}^k\right\| \le \left\|\boldsymbol{x}^{\natural} - \boldsymbol{v}^k\right\|,$$

which implies

$$\left\|\boldsymbol{u}^k_{\mathcal{T}_{k+1}} - \boldsymbol{v}^k_{\mathcal{T}_{k+1}}\right\| \le \left\|\boldsymbol{x}^{\natural}_{\mathcal{T}_{k+1}} - \boldsymbol{v}^k_{\mathcal{T}_{k+1}}\right\|,$$

because of the relation $\mathrm{supp}\left(\boldsymbol{u}^k\right) \subseteq \mathcal{T}_{k+1}$ and $\mathrm{supp}\left(\boldsymbol{x}^{\natural}\right) \subseteq \mathcal{T}_{k+1}$. Then, it follows from the triangle inequality and the inequality above that

$$
\begin{aligned}
\left\|\boldsymbol{u}^k - \boldsymbol{x}^{\natural}\right\| = \left\|\boldsymbol{u}^k_{\mathcal{T}_{k+1}} - \boldsymbol{x}^{\natural}_{\mathcal{T}_{k+1}}\right\| &= \left\|\boldsymbol{u}^k_{\mathcal{T}_{k+1}} - \boldsymbol{v}^k_{\mathcal{T}_{k+1}} + \boldsymbol{v}^k_{\mathcal{T}_{k+1}} - \boldsymbol{x}^{\natural}_{\mathcal{T}_{k+1}}\right\| \\
&\le \left\|\boldsymbol{u}^k_{\mathcal{T}_{k+1}} - \boldsymbol{v}^k_{\mathcal{T}_{k+1}}\right\| + \left\|\boldsymbol{v}^k_{\mathcal{T}_{k+1}} - \boldsymbol{x}^{\natural}_{\mathcal{T}_{k+1}}\right\| \quad (25) \\
&\le 2\left\|\boldsymbol{v}^k_{\mathcal{T}_{k+1}} - \boldsymbol{x}^{\natural}_{\mathcal{T}_{k+1}}\right\|.
\end{aligned}
$$

Define $\boldsymbol{z}^k := \boldsymbol{z} \odot \mathrm{sgn}\left(\boldsymbol{A}\boldsymbol{x}^k\right)$ with $\boldsymbol{z}$ as in (13). Using the definition of $\boldsymbol{v}^k$, a direct calculation yields

$$
\begin{aligned}
\left\|\boldsymbol{v}^k_{\mathcal{T}_{k+1}} - \boldsymbol{x}^{\natural}_{\mathcal{T}_{k+1}}\right\| &= \left\|\boldsymbol{x}^k_{\mathcal{T}_{k+1}} - \boldsymbol{x}^{\natural}_{\mathcal{T}_{k+1}} - \frac{\eta}{m} \boldsymbol{A}^T_{\mathcal{T}_{k+1}} \boldsymbol{A}\left(\boldsymbol{x}^k - \boldsymbol{x}^{\natural}\right) + \frac{\eta}{m} \boldsymbol{A}^T_{\mathcal{T}_{k+1}}\left(\boldsymbol{z}^k - \boldsymbol{A}\boldsymbol{x}^{\natural}\right)\right\| \\
&\le \underbrace{\left\|\frac{\eta}{m} \boldsymbol{A}^T_{\mathcal{T}_{k+1}}\left(\boldsymbol{z}^k - \boldsymbol{A}\boldsymbol{x}^{\natural}\right)\right\|}_{I_1} + \underbrace{\left\|\left(\boldsymbol{I} - \frac{\eta}{m} \boldsymbol{A}^T_{\mathcal{T}_{k+1}} \boldsymbol{A}_{\mathcal{T}_{k+1}}\right)\left(\boldsymbol{x}^k_{\mathcal{T}_{k+1}} - \boldsymbol{x}^{\natural}_{\mathcal{T}_{k+1}}\right)\right\|}_{I_2} \\
&\quad + \underbrace{\left\|\frac{\eta}{m} \boldsymbol{A}^T_{\mathcal{T}_{k+1}} \boldsymbol{A}_{\mathcal{T}_k \backslash \mathcal{T}_{k+1}}\left[\boldsymbol{x}^k - \boldsymbol{x}^{\natural}\right]_{\mathcal{T}_k \backslash \mathcal{T}_{k+1}}\right\|}_{I_3}.
\end{aligned}
$$

We will now proceed to estimate $I_1$, $I_2$, and $I_3$ sequentially.

For $I_1$: An application of Lemma B.4 yields that

$$\left\|\frac{\eta}{m} \boldsymbol{A}^T_{\mathcal{T}_{k+1}}\left(\boldsymbol{z}^k - \boldsymbol{A}\boldsymbol{x}^{\natural}\right)\right\| \le \eta\sqrt{C_\gamma\left(1 + \delta_{2s}\right)}\left\|\boldsymbol{x}^k - \boldsymbol{x}^{\natural}\right\|. \quad (26)$$

For $I_2$: Let $\eta \in \left(0, \frac{1}{1+\delta_{2s}}\right)$. It follows from (23) in Lemma B.10 as well as Weyl's inequality that

$$\left(1 - \eta(1 + \delta_{2s})\right)\left\|\boldsymbol{u}\right\| \le \left\|\left(\boldsymbol{I} - \frac{\eta}{m} \boldsymbol{A}^T_{\mathcal{T}_{k+1}} \boldsymbol{A}_{\mathcal{T}_{k+1}}\right)\boldsymbol{u}\right\| \le \left(1 - \eta(1 - \delta_{2s})\right)\left\|\boldsymbol{u}\right\|,$$

for any $\boldsymbol{u} \in \mathbb{R}^{|\mathcal{T}_{k+1}|}$, which deducts

$$\left\|\left(\boldsymbol{I} - \frac{\eta}{m} \boldsymbol{A}^T_{\mathcal{T}_{k+1}} \boldsymbol{A}_{\mathcal{T}_{k+1}}\right)\left(\boldsymbol{x}^k_{\mathcal{T}_{k+1}} - \boldsymbol{x}^{\natural}_{\mathcal{T}_{k+1}}\right)\right\| \le \left(1 - \eta(1 - \delta_{2s})\right)\left\|\boldsymbol{x}^k_{\mathcal{T}_{k+1}} - \boldsymbol{x}^{\natural}_{\mathcal{T}_{k+1}}\right\|.$$

For $I_3$: Eq.(22) in Lemma B.10 implies that

$$\left\|\frac{\eta}{m} \boldsymbol{A}^T_{\mathcal{T}_{k+1}} \boldsymbol{A}_{\mathcal{T}_k \backslash \mathcal{T}_{k+1}}\left[\boldsymbol{x}^k - \boldsymbol{x}^{\natural}\right]_{\mathcal{T}_k \backslash \mathcal{T}_{k+1}}\right\| \le \eta\delta_{3s}\left\|\left[\boldsymbol{x}^k - \boldsymbol{x}^{\natural}\right]_{\mathcal{T}_k \backslash \mathcal{T}_{k+1}}\right\|.$$

Combining all terms together, we obtain

$$
\begin{aligned}
\left\|\boldsymbol{v}^{k+1}_{\mathcal{T}_{k+1}} - \boldsymbol{x}^{\natural}_{\mathcal{T}_{k+1}}\right\| &\le I_1 + I_2 + I_3 \le \sqrt{2(I_1^2 + I_2^2)} + I_3 \\
&\le \sqrt{2}\max\left\{\eta\delta_{3s}, 1 - \eta\left(1 - \delta_{2s}\right)\right\}\left\|\boldsymbol{x}^k - \boldsymbol{x}^{\natural}\right\| + \eta\sqrt{C_\gamma\left(1 + \delta_{2s}\right)}\left\|\boldsymbol{x}^k - \boldsymbol{x}^{\natural}\right\| \\
&= \left(\sqrt{2}\max\left\{\eta\delta_{3s}, 1 - \eta\left(1 - \delta_{2s}\right)\right\} + \eta\sqrt{C_\gamma\left(1 + \delta_{2s}\right)}\right)\left\|\boldsymbol{x}^k - \boldsymbol{x}^{\natural}\right\|.
\end{aligned}
$$

We complete the proof by using (25). $\qquad\square$

### B.3 PROOF OF THEOREMS

We present the proofs of Theorems 3.1 and 3.2 in this subsection.

#### B.3.1 PROOF OF THEOREM 3.1

*Proof of Theorem 3.1.* In this proof, we consider the case where $\|\boldsymbol{x}^0 - \boldsymbol{x}^\natural\| \leq \|\boldsymbol{x}^0 + \boldsymbol{x}^\natural\|$. Consequently, the distance between the initial estimate and the true signal is given by $\mathrm{dist}(\boldsymbol{x}^0, \boldsymbol{x}^\natural) = \|\boldsymbol{x}^0 - \boldsymbol{x}^\natural\|$. Note that the case where $\|\boldsymbol{x}^0 + \boldsymbol{x}^\natural\| \leq \|\boldsymbol{x}^0 - \boldsymbol{x}^\natural\|$ can be addressed in a similar manner. For the purpose of this proof, we assume, without loss of generality, that the true signal has a unit norm, *i.e.*, $\|\boldsymbol{x}^\natural\| = 1$.

Let $\boldsymbol{x}^k$ represent the $k$-th iterate generated by Algorithm 1. Given an $s$-sparse estimate $\boldsymbol{x}^k$ with support $\mathcal{S}_k$, which is close to the target signal, *i.e.*, $\mathrm{dist}(\boldsymbol{x}^k, \boldsymbol{x}^\natural) \leq \gamma\|\boldsymbol{x}^\natural\|$. For any $0 \leq t \leq 1$, denote $\boldsymbol{x}(t) := \boldsymbol{x}^\natural + t(\boldsymbol{x}^k - \boldsymbol{x}^\natural)$. It is evident that $\mathrm{supp}(\boldsymbol{x}(t)) \subseteq \mathrm{supp}(\boldsymbol{x}^\natural) \cup \mathrm{supp}(\boldsymbol{x}^k)$, and the size of the support of $\boldsymbol{x}(t)$ is at most $2s$, *i.e.*, $|\mathrm{supp}(\boldsymbol{x}(t))| \leq 2s$. Furthermore, we obtain

$$\left\|\boldsymbol{x}^k - \boldsymbol{x}(t)\right\| \leq (1-t)\left\|\boldsymbol{x}^k - \boldsymbol{x}^\natural\right\| \leq \left\|\boldsymbol{x}^k - \boldsymbol{x}^\natural\right\|,$$

and

$$\left\|\boldsymbol{x}^k + \boldsymbol{x}(t)\right\| = \left\|(1+t)\boldsymbol{x}^k + (1-t)\boldsymbol{x}^\natural\right\| \leq (1+t)\left\|\boldsymbol{x}^k\right\| + (1-t)\left\|\boldsymbol{x}^\natural\right\|$$
$$\leq (1+t)(1+\gamma)\left\|\boldsymbol{x}^\natural\right\| + (1-t)\left\|\boldsymbol{x}^\natural\right\| \leq 2(\gamma+1)\left\|\boldsymbol{x}^\natural\right\|,$$

where the last inequality holds because $0 \leq t \leq 1$.

Assume events (20) and (14) hold, then by (15) in Lemma B.2 with $\boldsymbol{x} = \boldsymbol{x}^k, \boldsymbol{z} = \boldsymbol{x}(t), \mathcal{S} = \mathcal{S}_{k+1}, \mathcal{T} = \mathcal{S}_k \cup \mathcal{S}^\natural$ there holds

$$\left\|\nabla^2_{\mathcal{S}_{k+1},\mathcal{S}_k\cup\mathcal{S}^\natural} f_I\left(\boldsymbol{x}^k\right) - \nabla^2_{\mathcal{S}_{k+1},\mathcal{S}_k\cup\mathcal{S}^\natural} f_I(\boldsymbol{x}(t))\right\|$$
$$\leq \frac{3}{m}\left((3m)^{1/4} + (3s)^{1/2} + 2\sqrt{\log m}\right)^4 \left\|\boldsymbol{x}^k + \boldsymbol{x}(t)\right\| \left\|\boldsymbol{x}^k - \boldsymbol{x}(t)\right\| \tag{27}$$
$$\leq 10\left\|\boldsymbol{x}^k + \boldsymbol{x}(t)\right\| \left\|\boldsymbol{x}^k - \boldsymbol{x}(t)\right\|$$
$$\leq L_h\left\|\boldsymbol{x}^k - \boldsymbol{x}^\natural\right\|,$$

where $L_h := 20(\gamma+1)\|\boldsymbol{x}^\natural\|$. Also, Eq. (16) in Lemma B.3 with $\boldsymbol{x} = \boldsymbol{x}^k, \mathcal{S} = \mathcal{S}_{k+1}$ indicates that

$$\left\|\left(\nabla^2_{\mathcal{S}_{k+1},\mathcal{S}_{k+1}} f_I\left(\boldsymbol{x}^k\right)\right)^{-1}\right\| = \left(\lambda_{\min}\left(\nabla^2_{\mathcal{S}_{k+1},\mathcal{S}_{k+1}} f_I\left(\boldsymbol{x}^k\right)\right)\right)^{-1} \leq \frac{1}{l_1}. \tag{28}$$

Moreover, by the mean value theorem, one has

$$\nabla f_I\left(\boldsymbol{x}^k\right) - \nabla f_I\left(\boldsymbol{x}^\natural\right) = \int_0^1 \nabla^2 f_I\left(\boldsymbol{x}(t)\right)\left(\boldsymbol{x}^k - \boldsymbol{x}^\natural\right) dt. \tag{29}$$

We also have the following chain of equalities

$$\left\|\boldsymbol{x}^{k+1} - \boldsymbol{x}^\natural\right\| = \left[\left\|\boldsymbol{x}^{k+1}_{\mathcal{S}_{k+1}} - \boldsymbol{x}^\natural_{\mathcal{S}_{k+1}}\right\|^2 + \left\|\boldsymbol{x}^{k+1}_{\mathcal{S}^c_{k+1}} - \boldsymbol{x}^\natural_{\mathcal{S}^c_{k+1}}\right\|^2\right]^{1/2}$$
$$= \Big[\underbrace{\left\|\boldsymbol{x}^\natural_{\mathcal{S}^c_{k+1}}\right\|^2}_{I_1} + \underbrace{\left\|\boldsymbol{x}^k_{\mathcal{S}_{k+1}} - \boldsymbol{x}^\natural_{\mathcal{S}_{k+1}} + \boldsymbol{p}^k_{\mathcal{S}_{k+1}}\right\|^2}_{I_2}\Big]^{1/2}, \tag{30}$$

where $\boldsymbol{p}^k_{\mathcal{S}_{k+1}}$ is the search direction that can be obtained by (10).

We first bound the term $I_1$ in (30). Note that $\boldsymbol{x}^\natural_{\mathcal{S}^c_{k+1}}$ is a subvector of $\boldsymbol{x}^\natural - \boldsymbol{u}^k$. Based on this observation and applying Lemma B.5, we deduce that

$$I_1 = \left\|\boldsymbol{x}^\natural_{\mathcal{S}^c_{k+1}}\right\| \leq \left\|\boldsymbol{x}^\natural - \boldsymbol{u}^k\right\| \leq \zeta\left\|\boldsymbol{x}^k - \boldsymbol{x}^\natural\right\|, \tag{31}$$

where $\zeta = 2\left(\sqrt{2}\max\{\eta\delta_{3s}, 1 - \eta(1-\delta_{2s})\} + \eta\sqrt{C_\gamma(1+\delta_{2s})}\right)$ with $\eta < \frac{1}{1+\delta_{2s}}$.

We now proceed to bound the term $I_2$ in (30). By plugging the expression of $\boldsymbol{p}^k_{\mathcal{S}_{k+1}}$, we obtain that

$$I_2 = \left\| \left( \nabla^2_{\mathcal{S}_{k+1}, \mathcal{S}_{k+1}} f_I(\boldsymbol{x}^k) \right)^{-1} \left( \nabla^2_{\mathcal{S}_{k+1}, \mathcal{S}^c_{k+1}} f_I(\boldsymbol{x}^k) \boldsymbol{x}^k_{\mathcal{S}^c_{k+1}} - \nabla_{\mathcal{S}_{k+1}} f_I(\boldsymbol{x}^k) \right) + \boldsymbol{x}^k_{\mathcal{S}_{k+1}} - \boldsymbol{x}^\natural_{\mathcal{S}_{k+1}} \right\|. \tag{32}$$

We further can deduce that:

$$\begin{aligned}
I_2 &\leq \frac{1}{l_1} \left\| \nabla^2_{\mathcal{S}_{k+1}, \mathcal{S}^c_{k+1}} f_I(\boldsymbol{x}^k) \boldsymbol{x}^k_{\mathcal{S}^c_{k+1}} - \nabla_{\mathcal{S}_{k+1}} f_I(\boldsymbol{x}^k) + \nabla^2_{\mathcal{S}_{k+1}, \mathcal{S}_{k+1}} f_I(\boldsymbol{x}^k) \left( \boldsymbol{x}^k_{\mathcal{S}_{k+1}} - \boldsymbol{x}^\natural_{\mathcal{S}_{k+1}} \right) \right\| \\
&\leq \frac{1}{l_1} \left\| \int_0^1 \left[ \nabla^2_{\mathcal{S}_{k+1},:} f_I(\boldsymbol{x}^k) - \nabla^2_{\mathcal{S}_{k+1},:} f_I(\boldsymbol{x}(t)) \right] (\boldsymbol{x}^k - \boldsymbol{x}^\natural) dt \right\| + \frac{l_3}{l_1} \left\| \boldsymbol{x}^\natural_{\mathcal{S}^c_{k+1}} \right\| \\
&\leq \frac{1}{l_1} \int_0^1 \left\| \nabla^2_{\mathcal{S}_{k+1}, \mathcal{S}_k \cup \mathcal{S}^\natural} f_I(\boldsymbol{x}^k) - \nabla^2_{\mathcal{S}_{k+1}, \mathcal{S}_k \cup \mathcal{S}^\natural} f_I(\boldsymbol{x}(t)) \right\| \left\| \boldsymbol{x}^k - \boldsymbol{x}^\natural \right\| dt + \frac{\zeta l_3}{l_1} \left\| \boldsymbol{x}^k - \boldsymbol{x}^\natural \right\| \\
&\leq \frac{L_h}{l_1} \left\| \boldsymbol{x}^k - \boldsymbol{x}^\natural \right\|^2 \int_0^1 (1-t) dt + \frac{\zeta l_3}{l_1} \left\| \boldsymbol{x}^k - \boldsymbol{x}^\natural \right\| \\
&= \rho_1 \| \boldsymbol{x}^k - \boldsymbol{x}^\natural \|,
\end{aligned}$$

where the first inequality follows from (28), the second inequality is based on Lemma B.3 and (29) together with the fact that $\nabla f_I(\boldsymbol{x}^\natural) = \boldsymbol{0}$, and the third inequality is derived from (31), and the last inequality is obtained from (27). The equality includes $\rho_1$, defined as follows:

$$\begin{aligned}
\rho_1 &:= \frac{L_h}{2l_1} \| \boldsymbol{x}^k - \boldsymbol{x}^\natural \| + \frac{\zeta l_3}{l_1} \leq \frac{20(1+\gamma)\gamma}{2(2 - 2\delta - 10\gamma(2+\gamma))} + \frac{\zeta(2 + 2\delta + 10\gamma(2+\gamma))}{2 - 2\delta - 10\gamma(2+\gamma)} \\
&= \frac{2\zeta(1+\delta) + 10\gamma(1 + \gamma + \zeta(2+\gamma))}{2 - 2\delta - 10\gamma(2+\gamma)}.
\end{aligned}$$

By substituting the upper bounds of terms $I_1$ and $I_2$ into (30), we obtain:

$$\left\| \boldsymbol{x}^{k+1} - \boldsymbol{x}^\natural \right\| \leq \sqrt{\rho_1^2 + \zeta^2} \left\| \boldsymbol{x}^k - \boldsymbol{x}^\natural \right\|. \tag{33}$$

Let $\rho := \sqrt{\rho_1^2 + \zeta^2}$, then $\rho < 1$ can be ensured by properly choosing parameters. For example, when $\delta_{3s} \leq 0.05$ and $\delta = 0.001$, and set $\eta = 0.95$, then $\rho \leq 0.6 < 1$ provided that $\gamma \leq 0.01$. Therefore, $\|\boldsymbol{x}^{k+1} - \boldsymbol{x}^\natural\| \leq \rho \|\boldsymbol{x}^k - \boldsymbol{x}^\natural\| \leq \rho\gamma \|\boldsymbol{x}^\natural\|$ for some $\rho \in (0, 1)$.

Let $\mathcal{R}_k = \mathcal{S}_k \cup \mathcal{S}_{k+1} \cup \mathcal{S}^\natural$. Assume that the event (20) with $\mathcal{S} = \mathcal{R}_k$ holds for $k_1$ iterations. As stated in Theorem IV.1 of (Jagatap & Hegde, 2019), the initial guess $\boldsymbol{x}^0$ is guaranteed to satisfy $\mathrm{dist}(\boldsymbol{x}^0, \boldsymbol{x}^\natural) \leq \gamma \|\boldsymbol{x}^\natural\|$. By applying mathematical induction, we can show that for any integer $0 \leq k \leq k_1$, there exists a constant $\rho \in (0, 1)$ such that $\mathrm{dist}(\boldsymbol{x}^{k+1}, \boldsymbol{x}^\natural) \leq \rho \cdot \mathrm{dist}(\boldsymbol{x}^k, \boldsymbol{x}^\natural)$.

Let $K$ be the minimum integer such that it holds

$$\gamma \left\| \boldsymbol{x}^\natural \right\| \rho^K < x^\natural_{\min}. \tag{34}$$

We then assert that $\mathcal{S}^\natural \subseteq \mathcal{S}_k$ for all $k \geq K$. This is because if it were not the case, there would exist an index $i \in \mathcal{S}^\natural \backslash \mathcal{S}_k \neq \emptyset$, such that $\|\boldsymbol{x}^k - \boldsymbol{x}^\natural\| \geq |x^\natural_i| \geq x^\natural_{\min}$. However, this contradicts our assumption $\|\boldsymbol{x}^k - \boldsymbol{x}^\natural\| \leq \gamma \|\boldsymbol{x}^\natural\|\rho^k \leq \gamma \|\boldsymbol{x}^\natural\|\rho^K < x^\natural_{\min}$. Consequently, based on (34), we derive:

$$K = \left\lfloor \frac{\log(\gamma \|\boldsymbol{x}^\natural\|/x^\natural_{\min})}{\log(\rho^{-1})} \right\rfloor + 1 \leq C_a \log \left( \|\boldsymbol{x}^\natural\|/x^\natural_{\min} \right) + C_b.$$

Note that $\mathcal{S}_k = \mathcal{S}^\natural$ for all $k \geq K$ implies $\mathcal{R}_k = \mathcal{R}_{k+1}$ for all $k \geq K$. As a result, the probability of event (20) occurring for all $k \geq 0$ can be bounded by $1 - 4Km^{-2}$. To conclude, with probability at least $1 - (4K + C_c)m^{-1} - C_d e^{-C_e m/\log m}$, there holds

$$\mathrm{dist}(\boldsymbol{x}^{k+1}, \boldsymbol{x}^\natural) \leq \rho \cdot \mathrm{dist}(\boldsymbol{x}^k, \boldsymbol{x}^\natural).$$

with some $\rho \in (0, 1)$ provided $m \geq C_f s^2 \log(n/s)$. Moreover, for $k \geq K$, utilizing the result the result $\mathcal{S}_{k+1} = \mathcal{S}^\natural$ and consequently $\boldsymbol{x}^\natural_{\mathcal{S}^c_{k+1}} = \boldsymbol{0}$, we obtain:

$$\left\| \boldsymbol{x}^{k+1} - \boldsymbol{x}^\natural \right\| = \left[ \left\| \boldsymbol{x}^{k+1}_{\mathcal{S}_{k+1}} - \boldsymbol{x}^\natural_{\mathcal{S}_{k+1}} \right\|^2 + \left\| \boldsymbol{x}^{k+1}_{\mathcal{S}^c_{k+1}} - \boldsymbol{x}^\natural_{\mathcal{S}^c_{k+1}} \right\|^2 \right]^{1/2} = \left\| \boldsymbol{x}^k_{\mathcal{S}_{k+1}} - \boldsymbol{x}^\natural_{\mathcal{S}_{k+1}} + \boldsymbol{p}^k_{\mathcal{S}_{k+1}} \right\|.$$

We can further obtain

$$
\begin{aligned}
\left\| \boldsymbol{x}^{k+1} - \boldsymbol{x}^{\natural} \right\| &\leq \frac{1}{l_1} \left\| \nabla^2_{\mathcal{S}_{k+1},:} f_I(\boldsymbol{x}^k) \boldsymbol{x}^k - \nabla_{\mathcal{S}_{k+1}} f_I(\boldsymbol{x}^k) - \nabla^2_{\mathcal{S}_{k+1},\mathcal{S}_{k+1}} f_I(\boldsymbol{x}^k) \boldsymbol{x}^{\natural}_{\mathcal{S}_{k+1}} \right\| \\
&\leq \frac{1}{l_1} \left\| \nabla^2_{\mathcal{S}_{k+1},:} f_I(\boldsymbol{x}^k)(\boldsymbol{x}^k - \boldsymbol{x}^{\natural}) - \int_0^1 \nabla^2_{\mathcal{S}_{k+1},:} f_I(\boldsymbol{x}(t))(\boldsymbol{x}^k - \boldsymbol{x}^{\natural}) dt \right\| \\
&\leq \frac{1}{l_1} \left\| \int_0^1 \left[ \nabla^2_{\mathcal{S}_{k+1},:} f_I(\boldsymbol{x}^k) - \nabla^2_{\mathcal{S}_{k+1},:} f_I(\boldsymbol{x}(t)) \right] (\boldsymbol{x}^k - \boldsymbol{x}^{\natural}) dt \right\| \\
&\leq \frac{1}{l_1} \int_0^1 \left\| \nabla^2_{\mathcal{S}_{k+1},\mathcal{S}_k \cup \mathcal{S}^{\natural}} f_I(\boldsymbol{x}^k) - \nabla^2_{\mathcal{S}_{k+1},\mathcal{S}_k \cup \mathcal{S}^{\natural}} f_I(\boldsymbol{x}(t)) \right\| \left\| \boldsymbol{x}^k - \boldsymbol{x}^{\natural} \right\| dt \\
&\leq \frac{L_h}{l_1} \| \boldsymbol{x}^k - \boldsymbol{x}^{\natural} \|^2 \int_0^1 (1-t) dt \\
&= \frac{L_h}{2l_1} \| \boldsymbol{x}^k - \boldsymbol{x}^{\natural} \|^2,
\end{aligned}
$$

where the first inequality follows from (28) and (32), and the last inequality follows from (27). Consequently, the sequence $\{\boldsymbol{x}^k\}$ converges quadratically. $\qquad\square$

### B.3.2 PROOF OF THEOREM 3.2

*Proof of Theorem 3.2.* Owing to the independence between the sensing matrix and noise, along with the assumption $\mathbb{E}(\boldsymbol{\epsilon}) = \boldsymbol{0}$, Lemma B.11 remains valid in the noisy setting. Therefore, by applying Lemma B.1, if $m = \Omega(s \log n/s)$, then with overwhelming probability, it holds that

$$
\left\| \nabla^2_{\mathcal{S},\mathcal{S}} f_I \left( \boldsymbol{x}^{\natural} \right) - \mathbb{E} \left[ \nabla^2_{\mathcal{S},\mathcal{S}} f_I \left( \boldsymbol{x}^{\natural} \right) \right] \right\| \leq (\delta + \epsilon) \left\| \boldsymbol{x}^{\natural} \right\|^2.
$$

Under the assumptions in Lemma B.3, it follows that

$$
l_1' \| \boldsymbol{u} \| \leq \left\| \nabla^2_{\mathcal{S},\mathcal{S}} f_I(\boldsymbol{x}) \boldsymbol{u} \right\| \leq l_2' \| \boldsymbol{u} \|, \quad \forall \, \boldsymbol{u} \in \mathbb{R}^{|\mathcal{S}|}, \tag{35}
$$

and

$$
\left\| \nabla^2_{\mathcal{T},\mathcal{S}} f_I(\boldsymbol{x}) \right\| \leq l_3', \tag{36}
$$

where $l_1' = \left( 2 - 2\delta - 2\epsilon - 10\gamma(2+\gamma) \right) \| \boldsymbol{x}^{\natural} \|$, $l_2' = \left( 6 + 2\delta + 2\epsilon + 10\gamma(2+\gamma) \right) \| \boldsymbol{x}^{\natural} \|$, and $l_3' = \left( 2 + 2\delta + 2\epsilon + 10\gamma(2+\gamma) \right) \| \boldsymbol{x}^{\natural} \|$.

In the noisy case, $\{ \boldsymbol{z}^k \}_{k \geq 0}$ is given by

$$
\boldsymbol{z}^k = (|\boldsymbol{A}\boldsymbol{x}^{\natural}|^2 + \boldsymbol{\epsilon})^{\frac{1}{2}} \odot \operatorname{sgn}(\boldsymbol{A}\boldsymbol{x}^k).
$$

Then using the same argument as the proof of the inequality in Lemma B.4, we have

$$
\begin{aligned}
\left\| \frac{\eta}{m} \boldsymbol{A}^T_{\mathcal{T}_{k+1}} (\boldsymbol{z}^k - \boldsymbol{A}\boldsymbol{x}^{\natural}) \right\| &\leq \frac{\eta}{m} \left\| \boldsymbol{A}^T_{\mathcal{T}_{k+1}} \left( |\boldsymbol{A}\boldsymbol{x}^{\natural}| \odot \operatorname{sgn}(\boldsymbol{A}\boldsymbol{x}^k) - \boldsymbol{A}\boldsymbol{x}^{\natural} \right) \right\| \\
&\quad + \frac{C'\eta}{m} \left\| \boldsymbol{A}^T_{\mathcal{T}_{k+1}} \left( \boldsymbol{\epsilon} \odot \operatorname{sgn}(\boldsymbol{A}\boldsymbol{x}^k) \right) \right\| \\
&\leq \eta \sqrt{C_\gamma (1 + \delta_{2s})} \| \boldsymbol{x}^k - \boldsymbol{x}^{\natural} \| + \frac{C'\eta}{\sqrt{m}} \sqrt{1 + \delta_{2s}} \| \boldsymbol{\epsilon} \|,
\end{aligned} \tag{37}
$$

where the last inequality follows from Lemma B.4 and (21) in Lemma B.10. Then, we modify Lemma B.5 to estimate $\| \boldsymbol{u}^k - \boldsymbol{x}^{\natural} \|$ in the noisy case. All arguments in Lemma B.5 go except that the estimation of $I_1$ in (26) should be replaced by (37). Thus we obtain

$$
\left\| \boldsymbol{u}^k - \boldsymbol{x}^{\natural} \right\| \leq \zeta \left\| \boldsymbol{x}^k - \boldsymbol{x}^{\natural} \right\| + \frac{C'\eta}{\sqrt{m}} \sqrt{1 + \delta_{2s}} \| \boldsymbol{\epsilon} \|, \tag{38}
$$

where $\boldsymbol{u}^k, \eta$ are the same as those in Lemma B.5.

We now proceed to prove the convergence property for the noisy case, employing a similar argument as in the proof of Theorem 3.1. Notably, equality (30) remains valid in the presence of noise. As for the first term in (30), since $\boldsymbol{x}_{\mathcal{S}_{k+1}^c}^\natural$ is a subvector of $\boldsymbol{x}^\natural - \boldsymbol{u}^k$, it follows from (38) that

$$\left\|\boldsymbol{x}_{\mathcal{S}_{k+1}^c}^\natural\right\| \le \left\|\boldsymbol{x}^\natural - \boldsymbol{u}^k\right\| \le \zeta\|\boldsymbol{x}^k - \boldsymbol{x}^\natural\| + \frac{C'\eta}{\sqrt{m}}\sqrt{1 + \delta_{2s}}\|\boldsymbol{\epsilon}\|,$$

where $\zeta = 2\left(\sqrt{2}\max\{\eta\delta_{3s}, 1 - \eta(1 - \delta_{2s})\} + \eta\sqrt{C_\gamma(1 + \delta_{2s})}\right)$ with $\eta < 1/(1 + \delta_{2s})$.

Furthermore, the second term in (30) can now be estimated as

$$\left\|\boldsymbol{x}_{\mathcal{S}_{k+1}}^k - \boldsymbol{x}_{\mathcal{S}_{k+1}}^\natural + \boldsymbol{p}_{\mathcal{S}_{k+1}}^k\right\|$$

$$\le \frac{1}{l_1'}\left\|\nabla_{\mathcal{S}_{k+1},:}^2 f_I(\boldsymbol{x}^k)\boldsymbol{x}^k - \nabla_{\mathcal{S}_{k+1},:}^2 f_I(\boldsymbol{x}^k)\boldsymbol{x}^\natural + \nabla_{\mathcal{S}_{k+1},\mathcal{S}_{k+1}^c}^2 f_I(\boldsymbol{x}^k)\boldsymbol{x}_{\mathcal{S}_{k+1}^c}^\natural\right.$$
$$\left. - \nabla_{\mathcal{S}_{k+1}} f_I(\boldsymbol{x}^k) + \nabla_{\mathcal{S}_{k+1}} f_I(\boldsymbol{x}^\natural) + \frac{1}{m}\boldsymbol{A}_{\mathcal{S}_{k+1}}^T\left(\boldsymbol{\epsilon} \odot \left(\boldsymbol{A}\boldsymbol{x}^\natural\right)\right)\right\|$$

$$\le \frac{1}{l_1'}\left\|\nabla_{\mathcal{S}_{k+1},:}^2 f_I(\boldsymbol{x}^k)(\boldsymbol{x}^k - \boldsymbol{x}^\natural) - \int_0^1 \nabla_{\mathcal{S}_{k+1},:}^2 f_I(\boldsymbol{x}(t))(\boldsymbol{x}^k - \boldsymbol{x}^\natural)dt\right\|$$
$$+ \frac{1}{l_1'}\left\|\nabla_{\mathcal{S}_{k+1},\mathcal{S}_{k+1}^c}^2 f_I(\boldsymbol{x}^k)\boldsymbol{x}_{\mathcal{S}_{k+1}^c}^\natural\right\| + \frac{1}{l_1'm}\left\|\boldsymbol{A}_{\mathcal{S}_{k+1}}^T(\boldsymbol{\epsilon} \odot (\boldsymbol{A}\boldsymbol{x}^\natural))\right\|$$

$$\le \frac{1}{l_1'}\left\|\int_0^1 \left[\nabla_{\mathcal{S}_{k+1},:}^2 f_I(\boldsymbol{x}^k) - \nabla_{\mathcal{S}_{k+1},:}^2 f_I(\boldsymbol{x}(t))\right](\boldsymbol{x}^k - \boldsymbol{x}^\natural)dt\right\| + \frac{l_3'}{l_1'}\left\|\boldsymbol{x}_{\mathcal{S}_{k+1}^c}^\natural\right\| + \frac{\sqrt{1 + \delta_s}\|\boldsymbol{A}\boldsymbol{x}^\natural\|_\infty}{l_1'\sqrt{m}}\|\boldsymbol{\epsilon}\|$$

$$\le \frac{1}{l_1'}\int_0^1\left\|\nabla_{\mathcal{S}_{k+1},\mathcal{S}_k\cup\mathcal{S}^\natural}^2 f_I(\boldsymbol{x}^k) - \nabla_{\mathcal{S}_{k+1},\mathcal{S}_k\cup\mathcal{S}^\natural}^2 f_I(\boldsymbol{x}(t))\right\|\left\|\boldsymbol{x}^k - \boldsymbol{x}^\natural\right\|dt$$
$$+ \frac{\zeta l_3'}{l_1'}\left\|\boldsymbol{x}^k - \boldsymbol{x}^\natural\right\| + \frac{C'l_3'\eta\sqrt{1 + \delta_{2s}} + \sqrt{1 + \delta_s}\|\boldsymbol{A}\boldsymbol{x}^\natural\|_\infty}{l_1'\sqrt{m}}\|\boldsymbol{\epsilon}\|$$

$$\le \frac{L_h}{l_1'}\|\boldsymbol{x}^k - \boldsymbol{x}^\natural\|^2\int_0^1(1 - t)dt + \frac{\zeta l_3'}{l_1'}\left\|\boldsymbol{x}^k - \boldsymbol{x}^\natural\right\| + \frac{C'l_3'\eta\sqrt{1 + \delta_{2s}} + \sqrt{1 + \delta_s}\|\boldsymbol{A}\boldsymbol{x}^\natural\|_\infty}{l_1'\sqrt{m}}\|\boldsymbol{\epsilon}\|$$

$$= \rho_2\|\boldsymbol{x}^k - \boldsymbol{x}^\natural\| + \frac{C'l_3'\eta\sqrt{1 + \delta_{2s}} + \sqrt{1 + \delta_s}\|\boldsymbol{A}\boldsymbol{x}^\natural\|_\infty}{l_1'\sqrt{m}}\|\boldsymbol{\epsilon}\|, \tag{39}$$

where the first inequality follows from (35), together with the equality $\nabla f_I(\boldsymbol{x}^\natural) + \boldsymbol{A}^T\left(\boldsymbol{\epsilon} \odot \left(\boldsymbol{A}\boldsymbol{x}^\natural\right)\right) = \boldsymbol{0}$, and the third inequality is based on (36) and the inequality $\|\boldsymbol{a} \odot \boldsymbol{b}\| \le \|\boldsymbol{a}\|_\infty\|\boldsymbol{b}\|$ for any two vectors $\boldsymbol{a}, \boldsymbol{b}$. The last equality includes $\rho_2$, defined as follows:

$$\rho_2 := \frac{L_h}{2m_s'}\left\|\boldsymbol{x}^k - \boldsymbol{x}^\natural\right\| + \frac{\zeta h_s'}{m_s'} \le \frac{20(1 + \gamma)\gamma}{2(2 - 2\delta - 2\epsilon - 10\gamma(2 + \gamma))} + \frac{\zeta(2 + 2\delta + 2\epsilon + 10\gamma(2 + \gamma))}{2 - 2\delta - 2\epsilon - 10\gamma(2 + \gamma)}$$
$$= \frac{2\zeta(1 + \delta + \epsilon) + 10\gamma(1 + \gamma + \zeta(2 + \gamma))}{2 - 2\delta - 2\epsilon - 10\gamma(2 + \gamma)}.$$

Following from $(a^2 + b^2)^{1/2} \le a + b$ for $ab \ge 0$ and putting the two terms together yields

$$\left\|\boldsymbol{x}^{k+1} - \boldsymbol{x}^\natural\right\| \le \rho'\left\|\boldsymbol{x}^k - \boldsymbol{x}^\natural\right\| + \upsilon\|\boldsymbol{\epsilon}\|,$$

where $\rho' = \rho_2 + \zeta$ and $\upsilon = \frac{C'(l_1' + l_3')\eta\sqrt{1 + \delta_{2s}} + \sqrt{1 + \delta_s}\|\boldsymbol{A}\boldsymbol{x}^\natural\|_\infty}{l_1'\sqrt{m}}$. Noticing that

$$\frac{1}{\sqrt{m}}\left\|\boldsymbol{A}\boldsymbol{x}^\natural\right\|_\infty = \frac{1}{\sqrt{m}}\max_{i\in[m]}\left|\boldsymbol{a}_i^T\boldsymbol{x}^\natural\right| \le \frac{1}{\sqrt{m}}\max_{i\in[m],j\in\mathcal{S}^\natural}\left|a_{ij}\right| \cdot \left\|\boldsymbol{x}^\natural\right\| \le 3\sqrt{\frac{\log(ms)}{m}}\left\|\boldsymbol{x}^\natural\right\|$$

with probability at least $1 - (ms)^{-2}$. Consequently, $\frac{1}{\sqrt{m}}\|\boldsymbol{A}\boldsymbol{x}^\natural\|_\infty$ can be quite small. As a result, properly setting parameters can lead to $\upsilon \in (0, 1)$ and $\rho' \in (0, 1)$. $\qquad\square$

