# OpenReview forum: "A Fast and Provable Algorithm for Sparse Phase Retrieval"
_ICLR.cc/2024/Conference — ICLR 2024 poster_

### Official Review · Reviewer_5Ggq · 2023-10-19

**Soundness:** 3 good
**Presentation:** 3 good
**Contribution:** 3 good
**Rating:** 6
**Confidence:** 4

**Summary:**

This paper investigates the sparse phase retrieval problem, which seeks to reconstruct an unknown $s$-sparse vector $x$ from $m$ measurements $y_i=\langle a_i, x\rangle^2$. This problem has been extensively studied, and several first- and second-order methods have been developed. The main contribution of this paper is to propose a second-order algorithm for sparse phase retrieval. The cost per iteration is $\tilde{O}(s^4+s^2n^2)$, and the number of iterations is $\log \log (1/\epsilon)+\log (\|x\|/x_{\min})$. The algorithm also considers noisy measurements and proves a linear convergence rate. Additionally, the paper conducts several experiments in different parameter regimes and compares the new algorithm's performance with previous algorithms. The experimental results show that the new algorithm has significant advantages in efficiency, accuracy, and robustness.

**Strengths:**

The problem studied in this paper is of great significance in both theory and practice. The new fast algorithm proposed here can be utilized for a variety of real-world applications. Theoretically, the time complexity for each iteration of the algorithm matches the previous state-of-the-art first-order method, while the number of iterations improves on all previous first- and second-order methods. This is an essential theoretical improvement. Empirically, the proposed algorithm can be efficiently implemented and runs very fast and robustly. Furthermore, this paper is well-written, and most statements appear to be mathematically sound to me.

**Weaknesses:**

The super-linear convergence rate of the iterative method requires a warm start, meaning that it only holds after $K$ iterations. The main text of this paper does not provide enough intuition for why this initialization is necessary. Second, the initial point of the iterative method is generated via the sparse spectral initialization method by Jagatap and Hegde (2019). This makes the whole algorithm quite complicated. Third, it is unclear what are the technical differences between this paper and the papers by Cai et al. (2016) and Cai et al. (2022). It seems that the proofs in this paper rely on several technical lemmas from previous works.

**Questions:**

1. Page 2, bullet 2: what is |x|?
2. Page 2, bullet 2: “It is important to emphasize…” Note that the non-asymptotic convergence of Newton's methods has been studied in many optimization problems in theoretical computer science and optimization. It is indeed interesting to obtain an iteration bound for the sparse phase retrieval problem. But this emphasis seems to be unnecessary.
3. Page 3, Sec. 2.1: the formulation of the problem is unclear. For example, are the sensing vectors $a_i$ fixed or chosen by the algorithm?
4. Table 1: it would be better to add references to these methods.
5. Is there any lower bound on the computation complexity of this problem?
6. Page 4, Sec. 3.1: $O(n^3)$ can be replaced with $n^{\omega}$ using fast matrix multiplication since you are referring to the complexity.
7. Page 4: “We identify the set of free variables using one-step IHT”, IHT is not defined.
8. Eq. (4): is there a closed-form expression for the s-sparse hard thresholding? Does it just pick the s largest magnitude entries of the input vector?
9. Bottom of Page 4: is there any intuition as to why using f_A for free variable identification can improve the algorithm?
10. Thm. 3.1: “ There exists positive constant” -> “ exist”
11. Thm. 3.1: what is the convergence rate of the first K iterations?
12. Thm. 3.1: what’s the scale of $\rho$? Is it smaller than 1 or bigger than 1?
13. Thm 3.2: in this theorem, the convergence rate is characterized for the first K iteration. Why is this different from the noiseless case? Does it mean the algorithm is not robust in a long time?
14. Lem. B.2: Does $A_S$ mean the principal sub-matrix indexed by $S$? Or does it only include the rows in $S$? How tight is this lemma?
15. Lem. B.4: why $r=s$ and $s’$? Is it a typo here? Also, what is $\delta_{s+s’}$?
16. Lem. B.7: $\mathcal{T}$ is not defined here.
17. Eq. (29): it would be better to recall what is $p$ here.
18. Proof of Thm 3.1: It actually shows a linear convergence rate for the first K iterations. After that, the support of the ground-truth vector is fixed, and the quadratic convergence appears. It seems that in the fast-convergence phase, the algorithm uses Newton’s method to solve a general phase retrieval problem (without sparsity) of dimension $s$. Is the quadratic convergence rate known before for this problem?

---

> ### Author Response · Authors · 2023-11-20
>
> # Answers to Questions Part 1
>
> > 1. Page 2, bullet 2: what is $|x|$?
>
> **Reply:** Thank you for pointing out this. The term '$|x|$' was indeed a typo. It has been corrected to '$\Vert x \Vert$' in the revised manuscript, which represents the $\ell_2$-norm.
>
> > 2. Page 2, bullet 2: "It is important to emphasize…" Note that the non-asymptotic convergence of Newton's methods has been studied in many optimization problems in theoretical computer science and optimization. It is indeed interesting to obtain an iteration bound for the sparse phase retrieval problem. But this emphasis seems to be unnecessary.
>
> **Reply:** Thank you for your insightful comment. We agree that the particular emphasis on this aspect may not be necessary in our work. Accordingly, we have removed the mentioned sentences in the revised manuscript to improve the focus and clarity of our paper.
>
> > 3. Page 3, Sec. 2.1: the formulation of the problem is unclear. For example, are the sensing vectors $a_i$ fixed or chosen by the algorithm?
>
> **Reply:** Thank you for your comment. In phase retrieval problems, the sensing vectors $a_i$ are indeed given and fixed. We have clarified this point in the revised manuscript.
>
> > 4. Table 1: it would be better to add references to these methods.
>
> **Reply:** Thank you for your suggestion. Given the space limitations in the table, we have incorporated the references to these methods in the caption of the table in the revised manuscript.
>
> > 5. Is there any lower bound on the computation complexity of this problem?
>
> **Reply:** Thank you for your question. To the best of our knowledge, there is no established lower bound for the computation complexity of the sparse phase retrieval problem. However, we can provide some analysis for your interest.
>
> The computational complexity of solving the sparse phase retrieval problem is at least as much as the complexity of solving the compressive sensing problem, given that the former can be reduced to the latter if we assume the phases of the measurements to be known. Compressive sensing problems can typically be solved with a computational complexity of $O(s^2 n \log n)$, as demonstrated by algorithms like CoSaMP [D1] and SP [D2], where $n$ and $s$ are the dimension and sparsity of the signal, respectively. This complexity can be further reduced to $O(n \log^2 n)$ if the sampling matrix and its adjoint permit a fast matrix-vector multiply, and if the signals have power-law decaying entries [D1, D2].
>
> [D1] D. Needell and J.A. Tropp, "CoSaMP: Iterative signal recovery from incomplete and inaccurate samples," in Applied and Computational Harmonic Analysis, vol. 26, no. 3, pp. 301-321, 2009.
>
> [D2] W. Dai and O. Milenkovic, "Subspace Pursuit for Compressive Sensing Signal Reconstruction," in IEEE Transactions on Information Theory, vol. 55, no. 5, pp. 2230-2249, 2009.
>
> > 6. Page 4, Sec. 3.1: $O(n^3)$ can be replaced with $n^{\omega}$ using fast matrix multiplication since you are referring to the complexity.
>
> **Reply:** Thank you for your insightful comment. You are correct that the complexity of matrix multiplication can be represented as $n^{\omega}$, where $\omega < 3$ for fast matrix multiplication algorithms. In line with your suggestion, we have replaced $O(n^3)$ with $n^{\omega}$ in Section 3.1 of the revised manuscript, to more accurately represent the complexity in the context of fast matrix multiplication. We appreciate your valuable feedback, which has helped us improve the precision of our manuscript.
>
> > 7. Page 4: “We identify the set of free variables using one-step IHT”, IHT is not defined.
>
> **Reply:** We appreciate your pointing it out. 'IHT' stands for 'Iterative Hard Thresholding', a widely-used algorithm for solving sparse recovery problems. We have introduced the full term 'Iterative Hard Thresholding' at the first mention of its acronym 'IHT' in the revised manuscript.
>
> > 8. Eq. (4): is there a closed-form expression for the $s$-sparse hard thresholding? Does it just pick the s largest magnitude entries of the input vector?
>
> **Reply:** Thank you for your question. Yes, the $s$-sparse hard thresholding operation indeed involves picking the $s$ largest magnitude entries of the input vector and setting the rest to zero. We have clarified this in the revised manuscript.

---

> ### Author Response · Authors · 2023-11-20
>
> # Answers to Questions Part 2
>
> > 9. Bottom of Page 4: is there any intuition as to why using $f_A$ for free variable identification can improve the algorithm?
>
> **Reply:** Thank you for your insightful question. While our algorithm utilizes the intensity-based loss $f_I$ as the objective function, it might seem natural to use $f_I$ for identifying free variables in an algorithm. However, our numerical simulations revealed that employing the amplitude-based loss $f_A$ is more effective for this purpose. The potential reason is that $f_A$ often exhibits superior curvature around the truth compared to $f_I$, given that the curvature of $f_A$ behaves more similarly to that of a quadratic least-squares loss [E1,E2,E3]. As a result, the gradient of $f_A$ could potentially yield a more effective search direction than $f_I$. Furthermore, studies [E4,E5] have demonstrated that algorithms founded on $f_A$ consistently outperform those based on $f_I$ in numerical results. Given these observations, we have chosen to use the $f_A$ loss for free variable identification. We have included these insights in Section 3.1.3 of the revised manuscript.
>
> [E1] Y. Chi, Y. M. Lu and Y. Chen, "Nonconvex optimization meets low-rank matrix factorization: An overview," in IEEE Transactions on Signal Processing, vol. 67, no. 20, pp. 5239-5269, 2019.
>
> [E2] H. Zhang, Y. Liang, and Y. Chi, "A nonconvex approach for phase retrieval: Reshaped wirtinger flow and incremental algorithms," in Journal of Machine Learning Research, vol. 18, no. 141, pp. 1-35, 2017.
>
> [E3] G. Wang, G. B. Giannakis and Y. C. Eldar, "Solving systems of random quadratic equations via truncated amplitude flow," in IEEE Transactions on Information Theory, vol. 64, no. 2, pp. 773-794, 2018.
>
> [E4] L. Zhang, G. Wang, G. B. Giannakis and J. Chen, "Compressive phase retrieval via reweighted amplitude flow," in IEEE Transactions on Signal Processing, vol. 66, no. 19, pp. 5029-5040, 2018.
>
> [E5] J.-F. Cai, M. Huang, D. Li, and Y. Wang, "Solving phase retrieval with random initial guess is nearly as good as by spectral initialization," in Applied and Computational Harmonic Analysis, vol. 58, pp. 60-84, 2022.
>
> > 10. Thm. 3.1: “ There exists positive constant” -> “ exist”
>
> **Reply:** Thank you for your careful reading and the suggested correction. We have corrected this in the revised manuscript.
>
> > 11. Thm. 3.1: what is the convergence rate of the first $K$ iterations?
>
> **Reply:** Thank you for your question. The convergence rate is at least linear for the first $K$ iterations. This is demonstrated in the proof of Theorem 3.1. We have clarified this point in the paragraph following Theorem 3.1 in the revised manuscript.
>
> > 12. Thm. 3.1: what’s the scale of $\rho$? Is it smaller than 1 or bigger than 1?
>
> **Reply:** Thank you for your insightful question. The $\rho$ does not necessarily need to be smaller than 1 for establishing a quadratic convergence rate. The critical condition for achieving such a convergence rate is that $\rho$ remains a positive constant, that is, it does not grow with the iteration index. In addition, the convergence of the sequence is guaranteed, as demonstrated in the proof of Theorem 3.1, where we initially establish the convergence of the sequence to the ground truth. We appreciate your query and hope this clarification is helpful.
>
> > 13. Thm 3.2: in this theorem, the convergence rate is characterized for the first $K'$ iterations. Why is this different from the noiseless case? Does it mean the algorithm is not robust in a long time?
>
> **Reply:** Thank you for your insightful question. The proof of both Theorems 3.1 and 3.2 relies on the event (15) from Lemma B.2 holding with $\mathcal{S} = \mathcal{R}_k$ in each iteration, which occurs with a certain probability. To compute the total probability that the results in Theorems 3.1 and 3.2 hold, we need to union the probability that event (15) holds across all iterations.
>
> The key difference lies in the behavior of the set $\mathcal{R}_k$. In the proof of Theorem 3.1, we demonstrate that $\mathcal{R}_k$ remains fixed for any $k \geq K$, with $K$ specified in Theorem 3.1. Hence, we only need to consider the probability union for the first $K$ iterations. However, in the noisy scenario addressed by Theorem 3.2, we cannot guarantee a similar behavior of $\mathcal{R}_k$ remaining fixed. As a result, we introduce a general $K'$ and consider the probability union for the first $K'$ iterations.
>
> To clarify, Theorem 3.2 does not suggest that the established inequality will not hold when $k \geq K'$. Furthermore, the number of iterations required for our algorithm to converge is typically small, as it leverages second-order information when determining the search direction.
>
> We have added related comments in the paragraph following Theorem 3.2 in the revised manuscript for further clarification. We appreciate your query and hope this explanation is helpful.

---

> ### Author Response · Authors · 2023-11-20
>
> # Answers to Questions Part 3
>
> > 14. Lem. B.2: Does $A_S$ mean the principal sub-matrix indexed by $S$? Or does it only include the rows in $S$? How tight is this lemma?
>
> **Reply:** Thank you for raising these important points. In Lemma B.2, $A_\mathcal{S} \in \mathbb{R}^{m \times |\mathcal{S}|}$ represents the sub-matrix of $A \in \mathbb{R}^{m \times n}$, retaining only the columns indexed by $\mathcal{S}$. We have clarified this notation within the lemma itself in the revised manuscript.
>
> To further clarify, in our initial submission, $\nabla^2_{\mathcal{S}, \mathcal{T}} f (x) \in \mathbb{R}^{|\mathcal{S}| \times |\mathcal{T}|}$ denotes the sub-matrix of $\nabla^2 f (x) \in \mathbb{R}^{n \times n}$, with rows and columns indexed by $\mathcal{S}$ and $\mathcal{T}$, respectively. We further simplified $\nabla^2_{\mathcal{S}, \mathcal{S}} f (x)$ to $\nabla^2_{\mathcal{S}} f (x)$.
>
> To avoid confusion in our revised manuscript, we have chosen to remove this simplification, consistently using $\nabla^2_{\mathcal{S}, \mathcal{S}} f (x)$ (or occasionally, $\big[\nabla^2 f (x) \big]_{\mathcal{S}, \mathcal{S}}$) to indicate the principal sub-matrix of $\nabla^2 f (x)$ indexed by $\mathcal{S}$. We apologize for any confusion caused by our previous notation.
>
> As for the tightness of Lemma B.2, it is noteworthy that this result is not perfectly tight. This arises from the fact that this result is established by applying Slepian’s inequality and constructing a new Gaussian process to simplify estimation. However, the construction employed in this instance does not yield a tight upper bound. We appreciate your keen observations and insightful questions.
>
> > 15. Lem. B.4: why $r=s$ and $s’$? Is it a typo here? Also, what is $\delta_{s+s’}$?
>
> **Reply:** We apologize for any confusion. The terms $s$ and $s'$ in Lemma B.4 are indeed specific instances of a more general parameter $r$. To provide clarity, for a positive integer $r$ and a $\delta_r \in (0,1)$, we define an event such that, for any $r$-sparse vector $x \in \mathbb{R}^n$, the matrix $A$ satisfies the following inequality:
>
> $$ (1 - \delta_r) \Vert x \Vert^2 \leq \Vert \frac{1}{\sqrt{m}} A x \Vert^2 \leq (1 + \delta_r) \Vert x \Vert^2. $$
>
> In Lemma B.4, we consider the events where $r$ is specifically set to $s$ and $s'$, and we assume these events hold. These events are instances of the restricted isometry property (RIP) of order $r$, with $\delta_r$ as the associated restricted isometry constant.
>
> Furthermore, $\delta_{s+s’}$ is the restricted isometry constant corresponding to the RIP of order $s+s’$. Lemma B.1 establishes that Gaussian random matrices exhibit perfect RIP. To prevent any further confusion, we have refined the wording of Lemma B.4 in the revised manuscript.
>
> > 16. Lem. B.7: $\mathcal{T}_k$ is not defined here.
>
> **Reply:** Thank you for bringing this to our attention. Indeed, there was a typo here: $\mathcal{T}_k$ should have been $\mathcal{S}_k$. This set, $\mathcal{S}_k$, is generated by our algorithm in the first step, which involves identifying the set of free variables. We have updated Lemma B.7 accordingly in the revised manuscript. We appreciate your keen eye for detail.
>
> > 17. Eq. (29): it would be better to recall what is $\boldsymbol{p}$ here.
>
> **Reply:** Thank you for your valuable suggestion. Indeed, $\boldsymbol{p}$ denotes the search direction, as initially defined in Eq. (10). We have included a brief recall of this definition immediately following Eq. (29) in the revised manuscript. This should improve the readability of our paper.
>
> > 18. Proof of Thm 3.1: It actually shows a linear convergence rate for the first $K$ iterations. After that, the support of the ground-truth vector is fixed, and the quadratic convergence appears. It seems that in the fast-convergence phase, the algorithm uses Newton’s method to solve a general phase retrieval problem (without sparsity) of dimension $s$. Is the quadratic convergence rate known before for this problem?
>
> **Reply:** Thank you for your insightful comment and accurate understanding of our work. As you correctly noted, our proof of Theorem 3.1 shows a linear convergence rate for the first $K$ iterations, after which, with the ground-truth vector support fixed, it exhibits quadratic convergence. Our work is, to the best of our knowledge, the first to establish a quadratic convergence rate for the sparse phase retrieval problem.

---

> ### Author Response · Authors · 2023-11-21
>
> # Answers to Comments
>
> > 1. The main text of this paper does not provide enough intuition for why this initialization is necessary. The initial point of the iterative method is generated via the sparse spectral initialization method by Jagatap and Hegde (2019). This makes the whole algorithm quite complicated.
>
> **Reply:** We appreciate your comment. It's indeed true that all existing algorithms for solving the sparsity-constrained phase retrieval problem require a good initial guess to recover the ground truth. This is primarily due to the nonconvex nature of the problem, which can lead to multiple local minimizers. Our proposed method adopts a two-stage strategy: initialization and refinement.
>
> The purpose of the initialization stage is to generate an estimate that falls within a small-sized neighborhood around the true underlying signal. Our theoretical analysis demonstrates that, once the estimate enters this neighborhood, the iterative updates in the refinement stage will consistently stay within this neighborhood and be attracted towards the ground truth, exhibiting at least linear convergence and achieving quadratic convergence after $K$ iterations.
>
> While the sparse spectral initialization method we utilize may appear complex, it is notably effective and straightforward to implement. Furthermore, it would be intriguing to consider other initialization methods proposed in the literature, such as the sparse orthogonality-promoting initialization [F1] and diagonal thresholding initialization [F2].
>
> We have added relevant discussion in Section 3.2 of the revised manuscript.
>
> > 2. It is unclear what are the technical differences between this paper and the papers by Cai et al. (2016) and Cai et al. (2022). It seems that the proofs in this paper rely on several technical lemmas from previous works.
>
> **Reply:** While our paper relies on some technical lemmas from Cai et al. (2016) and Cai et al. (2022), referenced as [F2] and [F3], respectively, it addresses unique challenges and makes notable contributions. Firstly, our paper uniquely incorporates both intensity-based loss (as used in [F2]) and amplitude-based loss (as used in [F3]) in the development of our algorithm. Consequently, new convergence analysis techniques are required to tackle the technical challenges that arise when integrating these two types of losses into an algorithm.
>
> Secondly, the papers [F2, F3] and our paper develop algorithms based on different algorithmic frameworks. Our algorithm is grounded in the Newton projection method, whereas the algorithms in [F2] and [F3] are based on thresholded gradient descent and hard thresholding pursuit, respectively. This variation necessitates different techniques for theoretical convergence analysis. Notably, we establish quadratic convergence, a result not achieved in either [F2] or [F3].
>
> In summary, although our work builds upon previous studies, it stands apart in terms of the algorithmic framework and the comprehensive utilization of loss functions. These aspects underscore our unique contributions to the existing literature.
>
> [F1] G. Wang, L. Zhang, G. B. Giannakis, M. Akçakaya and J. Chen, "Sparse phase retrieval via truncated amplitude flow," in IEEE Transactions on Signal Processing, vol. 66, no. 2, pp. 479-491, 2018.
>
> [F2] T. T. Cai, X. Li, and Z. Ma, "Optimal rates of convergence for noisy sparse phase retrieval via thresholded wirtinger flow," in Annals of Statistics, vol. 44, no. 5, pp. 2221–2251, 2016.
>
> [F3] J.-F. Cai, J. Li, X. Lu, and J. You, "Sparse signal recovery from phaseless measurements via hard thresholding pursuit," in Applied and Computational Harmonic Analysis, vol. 56, pp. 367–390, 2022.

---

> ### Comment · Reviewer_5Ggq · 2023-11-22
>
> Thanks for your detailed response. I’ll keep my score unchanged.

---

> > ### Author Response · Authors · 2023-11-23
> >
> > Thank you for your time and for reviewing our responses. We value your engagement and the opportunity it has provided us to elucidate various aspects of our work.

---

### Official Review · Reviewer_qMEw · 2023-10-29

**Soundness:** 3 good
**Presentation:** 3 good
**Contribution:** 3 good
**Rating:** 8
**Confidence:** 4

**Summary:**

This paper proposes a new (second order) algorithm for the sparse phase retrieval problem, which is based on Newton projection. The algorithm is proven to converge to the ground truth at a quadratic convergence rate. This paper also provides various experiments to demonstrate its performance against SOTA algorithm (e.g., with/without noise and phase transition plots).

**Strengths:**

1. This paper is easy to follow for audience with phase retrieval background.
2. Theoretical results for both with/without noise are provided.
3. Experiments are sufficient, and demonstrate its performance against SOTA algorithms. I really like the phase transition plot, which will be asked to do so if it is not provided.

**Weaknesses:**

Although it is easy to follow for audience with phase retrieval background, I suggest authors to provide more details such as intuitions and figures of geometry to improve the readability for a broader audience. A good example will be ``Yuxin Chen and Emmanuel J Candès. Solving random quadratic systems of equations is nearly as easy as solving linear systems. Communications on pure and applied mathematics, 70(5):822–883, 2017.'' I will raise my rating if it could be realized.

**Questions:**

N/A

---

> ### Author Response · Authors · 2023-11-20
>
> # Answers to Questions
>
> > Although it is easy to follow for audience with phase retrieval background, I suggest authors to provide more details such as intuitions and figures of geometry to improve the readability for a broader audience. A good example will be ``Yuxin Chen and Emmanuel J Candès. Solving random quadratic systems of equations is nearly as easy as solving linear systems. Communications on pure and applied mathematics, 70(5):822–883, 2017.'' I will raise my rating if it could be realized.
>
> **Reply:** We sincerely appreciate the insightful comments. Your suggestion to develop intuitive explanations, much like those seen in [C1], is deeply valued. Indeed, such clear and accessible understanding is crucial for communicating our work to a wider audience.
>
> In response to your suggestions, we have revised our manuscript to provide more intuitive explanations of our algorithm, with a particular focus on the utilization of two types of loss functions: the intensity-based loss and the amplitude-based loss.
>
> The intensity-based loss $f_I$ serves as our primary objective function. Its smooth nature enables us to compute the Hessian and construct the Newton direction. Interestingly, while one might intuitively use the same loss function to determine which variables to update in each iteration, our numerical simulations suggest that the amplitude-based loss $f_A$ is more effective for this purpose.
>
> One potential reason is the superior curvature exhibited by $f_A$ around the true solution, which often behaves more similarly to that of a quadratic least-squares loss [C2, C3, C4]]. This indicates that the gradient of $f_A$ could offer a more effective search direction than that of $f_I$. Furthermore, studies [C5, C6]] have demonstrated that algorithms founded on $f_A$ consistently outperform those based on $f_I$ in numerical results.
>
> The simultaneous use of these two loss functions also improves the robustness of our algorithm. This strategy allows the algorithm to leverage the strengths of each loss function, increasing its adaptability and efficiency. It is akin to having two different skillsets, enabling us to navigate through the solution space with enhanced resilience.
>
> **We have included these intuitive explanations in a new subsection (Section 3.1.3) in the revised manuscript.** We believe these insights will help clarify the methodology and effectiveness of our algorithm. Your constructive feedback has been instrumental in improving the accessibility and depth of our manuscript, making it more comprehensible to a wider audience.
>
> [C1] Y. Chen and E. J. Candès, "Solving random quadratic systems of equations is nearly as easy as solving linear systems," in Communications on Pure and Applied Mathematics, vol. 70, no. 5, pp. 822–883, 2017.
>
> [C2] Y. Chi, Y. M. Lu and Y. Chen, "Nonconvex optimization meets low-rank matrix factorization: An overview," in IEEE Transactions on Signal Processing, vol. 67, no. 20, pp. 5239-5269, 2019.
>
> [C3] H. Zhang, Y. Liang, and Y. Chi, "A nonconvex approach for phase retrieval: Reshaped wirtinger flow and incremental algorithms," in Journal of Machine Learning Research, vol. 18, no. 141, pp. 1-35, 2017.
>
> [C4] G. Wang, G. B. Giannakis and Y. C. Eldar, "Solving systems of random quadratic equations via truncated amplitude flow," in IEEE Transactions on Information Theory, vol. 64, no. 2, pp. 773-794, 2018.
>
> [C5] L. Zhang, G. Wang, G. B. Giannakis and J. Chen, "Compressive phase retrieval via reweighted amplitude flow," in IEEE Transactions on Signal Processing, vol. 66, no. 19, pp. 5029-5040, 2018.
>
> [C6] J.-F. Cai, M. Huang, D. Li, and Y. Wang, "Solving phase retrieval with random initial guess is nearly as good as by spectral initialization," in Applied and Computational Harmonic Analysis, vol. 58, pp. 60-84, 2022.

---

> > ### Comment · Reviewer_qMEw · 2023-11-22
> >
> > Thanks for your response, and my concern is addressed. The rating is adjusted accordingly.

---

> > > ### Author Response · Authors · 2023-11-22
> > >
> > > We sincerely thank you for your prompt response and the time you've devoted. Your constructive feedback is greatly appreciated.

---

### Official Review · Reviewer_M25C · 2023-10-31

**Soundness:** 3 good
**Presentation:** 3 good
**Contribution:** 3 good
**Rating:** 6
**Confidence:** 4

**Summary:**

This paper proposes a second-order Newton-type algorithm for solving the sparse phase retrieval problems. It is shown that the algorithm enjoys a quadratic convergence rate with similar sample complexies relative to existing algorithms. Numerical tests verify the theoretical findings.

**Strengths:**

+ The design of the algorithm is interesting and delicate which is different than the existing sparse PR algorithms based on hard/soft thresholding.
+ The paper is easy to follow.
+ Extensive convincing numerical results.

**Weaknesses:**

- The first stage of finding the initialization is not reinvented.
- Known sparsity level s a priori.
- When noise is considered, the convergence rate recovers that of the first-order algorithms. In practice, all measurements are noisy. Please explain what is the advantage of the proposed algorithm for practical settings then.

**Questions:**

In different papers of sparse PR, SPARTA performs better and faster than e.g., ThWF; see e.g., the paper CoPRAM. It is strange why SPARTA is the worst performing method in almost all tests provided here. Please explain and replicate an experimental setting in SPARAT or CoPRAM paper using all methods.

---

> ### Author Response · Authors · 2023-11-20
>
> # Answers to Questions
>
> > 1. In different papers of sparse PR, SPARTA performs better and faster than e.g., ThWF; see e.g., the paper CoPRAM. It is strange why SPARTA is the worst performing method in almost all tests provided here.
>
> **Reply:** We appreciate your keen observation regarding the discrepancy in the phase transition performance of SPARTA and ThWF, as compared with the CoPRAM paper [A1]. Upon careful review, we identified that this divergence stems from the particular parameter settings employed in the methods.
>
> In our initial phase transition experiments depicted in Figure 2, we used the same parameter settings as those in Figure 1, where we had fine-tuned the parameters for each method. We acknowledge that these parameters could be further optimized for the experiments in Figure 2, and that the settings utilized in paper [A1] are indeed more appropriate.
>
> Prompted by this insight, **we have updated the results in Figure 2 of our paper to align with the parameter settings from [A1]**. We noticed a substantial improvement in the performance of SPARTA relative to ThWF in Figure 2(a), where the sparsity level $s = 25$. As the sparsity level increased to $s = 50$ in Figure 2(b), the performance gap between the two methods narrowed.
>
> We have incorporated these updated results for Figure 2 in the revised manuscript. It is important to emphasize that these revised results do not change the conclusions drawn regarding our proposed method. We express our sincere appreciation to the reviewer for illuminating this issue. Your feedback has been instrumental in enhancing the accuracy and robustness of our research.
>
> > 2. Please explain and replicate an experimental setting in SPARAT or CoPRAM paper using all methods.
>
> **Reply:** We are grateful for your insightful recommendation. Following your advice, we have replicated the experiments illustrated in Figure 2 of the CoPRAM paper [A1]. To be precise, we used the same experimental parameters for signal generation and kept the same parameter settings for CoPRAM, ThWF, and SPARTA. The only difference lies in the number of trials and the criterion for a successful recovery, adjusted to ensure consistency with our settings throughout the paper. We increased the number of trials from 50 to 200 and considered a recovery successful if the relative error was less than $10^{-3}$, rather than $0.05$.
>
> **We have integrated these experimental results into Section A.2 of the revised manuscript.** In alignment with the observations in Figure 2 of our paper and Figure 2 of [A1], we observed that SPARTA's phase transition performance significantly surpasses that of ThWF when the sparsity level is set to $s = 20$. As we elevated the sparsity level to $s = 30$, the performance gap remained, albeit slightly reduced. We extend our sincere appreciation to the reviewer for your invaluable input.
>
> [A1] G. Jagatap and C. Hegde, "Sample-efficient algorithms for recovering structured signals rrom magnitude-only measurements," in IEEE Transactions on Information Theory, vol. 65, no. 7, pp. 4434-4456, 2019.

---

> ### Author Response · Authors · 2023-11-20
>
> # Answers to Comments
>
> > 1. The first stage of finding the initialization is not reinvented.
>
> **Reply:** We appreciate your insightful comment. The primary focus of our work is on the refinement stage, where we introduce a novel second-order algorithm.
>
> In the initialization stage, we have adopted a sparse spectral initialization technique. However, we openly acknowledge and appreciate the diversity of other well-established initialization methods for sparse phase retrieval, such as sparse orthogonality-promoting initialization [B1] and diagonal thresholding initialization [B2].
>
> In consideration of your feedback, we have incorporated relevant discussions into Section 3.2 of the revised manuscript, highlighting the importance and diversity of initialization methods in sparse phase retrieval.
>
> > 2. Known sparsity level $s$ a priori.
>
> **Reply:** We appreciate your comment. Indeed, our work operates under the assumption that the sparsity level is known a priori. However, as evidenced by the experiments detailed in Table 4 of the revised submission (i.e., Table 3 of the initial submission), this assumption does not hinder the effectiveness of our proposed algorithm. The experiments demonstrate that our algorithm, along with other compared methods, exhibits notable robustness to variations in input sparsity. This illustrates its applicability and resilience in real-world scenarios where exact sparsity may not always be known.
>
> > 3. When noise is considered, the convergence rate recovers that of the first-order algorithms. In practice, all measurements are noisy. Please explain what is the advantage of the proposed algorithm for practical settings then.
>
> **Reply:** We appreciate your insightful comment. You are correct in noting that our theoretical analysis shows a linear convergence rate for our algorithm in the presence of noise – a rate that aligns with those of first-order algorithms. However, our proposed method offers a unique advantage: it incorporates second-order information when determining the search direction, a feature not typically found in first-order algorithms.
>
> As a result of this, we have observed in our experiments that our algorithm converges more quickly than the compared methods, even under noisy conditions. In response to your valuable feedback, we have enhanced our discussion on this particular aspect in the revised manuscript, especially in the paragraph following Theorem 3.2.
>
> [B1] G. Wang, L. Zhang, G. B. Giannakis, M. Akçakaya and J. Chen, "Sparse phase retrieval via truncated amplitude flow," in IEEE Transactions on Signal Processing, vol. 66, no. 2, pp. 479-491, 2018.
>
> [B2] T. T. Cai, X. Li, and Z. Ma, "Optimal rates of convergence for noisy sparse phase retrieval via thresholded wirtinger flow," in Annals of Statistics, vol. 44, no. 5, pp. 2221–2251, 2016.

---

### Official Review · Reviewer_s5s7 · 2023-11-01

**Soundness:** 3 good
**Presentation:** 3 good
**Contribution:** 3 good
**Rating:** 8
**Confidence:** 3

**Summary:**

In this paper, the authors address the sparse phase retrieval problem, which focuses on the recovery of a sparse signal from a limited number of phaseless measurements. The authors propose an efficient second-order algorithm based on Newton's projection, which maintains the same per-iteration computational complexity as popular first-order methods. The paper's primary focus is on the sparse phase retrieval problem, where the goal is to recover a sparse signal from phaseless measurements. The proposed algorithm is theoretically guaranteed to converge to the ground truth at a quadratic convergence rate after some upper-bounded iterations. Numerical experiments demonstrate that their algorithm outperforms state-of-the-art methods in terms of achieving a significantly faster convergence rate.

**Strengths:**

The authors have introduced an efficient second-order algorithm rooted in Newton projection that maintains an equivalent per-iteration computational complexity to widely used first-order methods. The algorithm is accompanied by a theoretical guarantee, ensuring a quadratic convergence rate when converging to the ground truth. Their algorithm's superior performance is substantiated through numerical experiments, demonstrating a notably faster convergence rate compared to existing state-of-the-art methods.

**Weaknesses:**

The experimental section could provide a more comprehensive perspective on the algorithm's efficiency by considering different dimensions. For example, it would be beneficial to explore why the experiments were consistently conducted with fixed dimensions, such as signal dimension (n) at 5000 and sample size (m) at 3000. This choice of fixed dimensions raises questions about the scalability of efficiency when the dimension varies.

**Questions:**

See weakness.

---

> ### Author Response · Authors · 2023-11-20
>
> # Answers to Questions
>
> > The experimental section could provide a more comprehensive perspective on the algorithm's efficiency by considering different dimensions. For example, it would be beneficial to explore why the experiments were consistently conducted with fixed dimensions, such as signal dimension (n) at 5000 and sample size (m) at 3000. This choice of fixed dimensions raises questions about the scalability of efficiency when the dimension varies.
>
> **Reply:** We appreciate your insightful comments regarding the dimensions used in our experiments. In response to your suggestion, we carried out additional experiments to explore the scalability of our algorithm with varying dimensions. Specifically, we expanded the range of signal dimensions from $10000$ to $50000$ and varied the sample size ratio $(m/n)$ from 0.3 to 0.7.
>
> The table below presents the running time comparison (in seconds) for our proposed algorithm when recovering signals with various dimensions and sample size ratios. The reported results represent the average of 200 independent trial runs.
>
> | Dimension $n$  |  $1 \times 10^5 $ |  $1.5 \times 10^5 $  |  $2 \times 10^5 $  |  $2.5 \times 10^5 $  |   $3 \times 10^5 $  |  $3.5 \times 10^5 $  |    $4 \times 10^5 $  |  $4.5 \times 10^5 $  |  $5 \times 10^5 $   |
> |------------------------|-----|-----|---|-----|---|-----|---|-----|---|
> | $m/n = 0.3$ | 0.212 | 0.420 | 0.458 | 0.675 | 0.987 | 1.214 | 1.472 | 1.854 | 2.147 |
> |  $m/n = 0.5$  | 0.242 | 0.497 | 0.585 | 0.912 | 1.382 | 1.788 | 2.237 | 2.709 | 3.022 |
> |  $m/n = 0.7$  | 0.278 | 0.545 | 0.833 | 1.179 | 1.783 | 2.224 | 2.789 | 3.581 | 4.252 |
>
> These additional results provide a more comprehensive perspective on our algorithm's efficiency and scalability, addressing your concerns about the fixed dimensions in our initial experiments. We believe that these additional tests offer a clearer understanding of the scalability of our proposed algorithm.
>
> **We have integrated these results into Section A.1 of the revised manuscript, which we believe has significantly enhanced its quality, thanks to your valuable feedback.** Once again, we express our sincere gratitude for your constructive comments.

---

> > ### Comment · Reviewer_s5s7 · 2023-11-22
> >
> > I appreciate your efforts in addressing my concerns.

---

> > > ### Author Response · Authors · 2023-11-22
> > >
> > > Thank you for your acknowledgment. We greatly appreciate your constructive feedback.

---

### Meta-Review · Area_Chair_hBRh · 2023-12-15

**Metareview:**

This paper proposes a second-order method for the sparse phase retrieval problem and establishes its convergence rate. The reviewers found the contribution interesting and solid. However, the section on numerical experiments should be strengthened.

**Justification For Why Not Higher Score:**

The paper tests the proposed method on problem instances that are generated from an idealized model. Thus, it does not provide convincing evidence that the method can work on real datasets.

**Justification For Why Not Lower Score:**

The paper contains good theoretical contributions to the algorithmic aspects of the sparse phase retrieval problem.

---

### Decision · Program_Chairs · 2024-01-16

Accept (poster)